# Characteristics of bacterial community in cloud water at Mt. Tai: similarity and disparity under polluted and non-polluted cloud episodes

Min Wei [1], Caihong Xu[1], Jianmin Chen[1,2,*], Chao Zhu[1], Jiarong Li[1], Ganglin Lv [1]

[1] Environment Research Institute, School of Environmental Science and Engineering, Shandong University, Ji'nan 250100, China

[2] Shanghai Key Laboratory of Atmospheric Particle Pollution and Prevention (LAP), Fudan Tyndall Centre, Department of Environmental Science & Engineering, Fudan University, Shanghai 200433, China

*Correspondence to*: JM.Chen (jmchen@sdu.edu.cn)

**Abstract:**

Bacteria, widely distributed in atmospheric bioaerosols, are indispensable components in clouds and play an important role in atmospheric hydrological cycle. However, limited acknowledge is acquired about bacterial community structure and function, especially for the increasing air pollution events in North China Plain. Here we presented a comprehensive characterization of bacterial community composition, function, variation and environmental influence for the cloud water collected at Mt. Tai from 24 Jul to 23 Aug 2014. Using the Miseq 16S rRNA gene sequencing, the facts that cloud water harbored a highly diverse bacterial community and the predominant phyla of Proteobacteria, Bacteroidetes, Cyanobacteria and Firmicutes were investigated. The presence of bacterial taxa survived in low temperature, radiation and poor nutrients conditions were encountered in cloud water, suggesting well adaption to extreme environment. Bacterial gene functions predicted from 16S rRNA gene using the Phylogenetic Investigation of Communities by Reconstruction of Unobserved States (PICRUSt) suggested the pathways relating to metabolism and disease infections are significantly correlated to the predominant genera. The abundant genera *Acinetobacter*, *Stenotrophomonas*, *Pseudomonas*, and *Empedobacter*

originated from a wide range of habitat included cloud condensation nuclei and ice nuclei active species, opportunistic pathogenic and functional species, demonstrating the bacterial ecological and healthy importance in cloud water should be concerned. Clustering analysis including hierarchical cluster (Hcluster) and principal coordinate analysis (PCoA) indicated a significant disparity between polluted and non-polluted samples. Linear discriminant analysis effect size (LEfSe) demonstrated that the polluted cloud samples were enriched with potential pathogens. The non-polluted samples had more diverse ecological function groups. Community structure discrepant performed by redundancy analysis (RDA) indicated that major ions in cloud water and $PM_{2.5}$ have negative impact on bacteria, playing vital role in shaping microbial community structure. Major ions may provide available nutrition for bacteria and have direct influence on bacterial community, whereas $PM_{2.5}$ in air had an indirect impact on bacterial community structure. During wet deposition, soluble particulate matter dissolved in water droplets and resulted in the elevated concentration in cloud water. $PM_{2.5}$ was possibly associated with different origins and pathways of air mass using source tracking by the backward trajectory and wind analysis, mainly related to the long-term transport. This work furthered our understanding of bacterial ecological characteristics in the atmospheric aqueous phase, highlighted the potential influence of environmental variables on bacterial community over cloud process. It may provide fundamental acquaintance of bacterial community response in cloud water under increasing pollution stress.

**Key words**:cloud water, 16r RNA gene, function prediction, major ions, $PM_{2.5}$

## 1. Introduction

Cloud is the aerosol system composed of tiny droplets suspended in the atmosphere. In the atmosphere, pollutants attached to particles could be dissolved or incorporated into cloud droplets, which may induce complex impacts on environment security and human health. Over the past decades, studies on cloud water have mainly focused on the physicochemical properties (Aikawa et al., 2001; Boris et al., 2015; Fernández-González et al., 2014). Recently, with the in-depth understanding of the characteristics of cloud, studies on bioaerosols in clouds have been in the ascendant.

Studies have showed that living microorganisms, including bacteria, fungi and yeasts, are present in clouds (Burrows et al., 2009). As the first study on biological particles in fog/cloud water, Fuzzi et al (1997) suggest the bacterial replication in foggy days. Afterwards, with the development of detection techniques, microorganisms in fog/cloud water have been systematically studied (Amato et al., 2007c; Delort et al., 2010; Vaïtilingom et al., 2012). Combined with the field investigations and lab experiments, diverse bacterial communities are retrieved, and the bacterial metabolism active in cloud water are further demonstrated. In atmospheric aqueous phase, microorganisms can act as cloud condensation nuclei (CCN) and ice nuclei (IN), which have potential impact on cloud formation and precipitation processes (Amato et al., 2015; Bauer et al., 2003; Mortazavi et al., 2015). Moreover, microorganisms in cloud water are available to metabolize organic carbon compounds (degrading organic acids, formate, acetate, lactate, succinate) and associate with carbon and nitrogen recycling (Amato et al., 2007a; Hill et al., 2007; Vaïtilingom et al., 2010). They can also influence photochemical chemical reactions (Vaïtilingom et al., 2013) and participate in a series of complex and diverse biochemical metabolic activities.

Cloud occurrence is a complex process, in contaminated areas, cloud typically contains numerous pollutants, e.g., sulfate and nitrate ions, organic carbon compounds, and bacteria (Badarinath et al., 2007; Després et al., 2012; Fernández-González et al., 2014; Mohan & Payra, 2009). As an intensive agricultural and economic region in China, the North China Plain has been suffering serious air pollution in recent years, e.g., the severe fog and haze pollution in Beijing, Ji'nan in January 2013 (Huang et al., 2014; Wang et al., 2014). Mt. Tai ($36^o18'$ N, $117^o13'$ E, and 1534 m a.s.l), the highest

mountain in North China Plain, is frequently attacked by cloud episodes (Guo et al., 2012a; Liu et al., 2012). Emissions and resuspension of bacteria by wind erosion or splashing water from various terrestrial environments into atmosphere recruit diverse airborne bacteria, which possibly involve opportunistic and functional bacteria. During cloud process, these bacteria attached to particles or incorporated in cloud droplets will be deposited back to land via dry or wet deposition. Accumulating literatures have indicated that bacteria in cloud/fog water droplets have potential effect on diversity and function of atmospheric and terrestrial ecosystems (Delort et al., 2010; Vaïtilingom et al., 2013), even induce health risks through microbial pathogens dispersion (Vaïtilingom et al., 2012). Previous literatures have studied the bacterial community in rain or snow (Cho & Jang, 2014; Mortazavi et al., 2015). They also focus on the bacteria associated with CNN/IN, potential pathogens and biochemical reactions. Therefore, to evaluate the potential ecological functional bacteria in cloud water have been an urgent issue, especially for the polluted cloud episodes.

It is noteworthy that atmospheric microorganisms are subject to a wide range of environmental condition including the meteorological factors and aerosols physiochemical composition (Womack et al., 2010). Community structure and function are closely related to the environmental characteristics in atmosphere and geomorphic characteristics (Dong et al., 2016; Gao et al., 2016). For instance, studies about inhalable bioaerosols in particulate matter suggest environmental parameters including temperature, relative humidity, $PM_{10}$, $PM_{2.5}$ and particle size have significant impact on the composition and dynamic of microbial communities (Adhikari et al., 2006; Bowers et al., 2013). However, due to the paucity of detailed and comprehensive studies of atmospheric bacterial composition, the understanding of the dynamic of bacterial community remains incomplete. During polluted cloud process, how bacterial community varied and which environmental factor play decisive role in shaping bacterial community structure are still scarcely studied.

In the present work, typical cloud episodes under polluted and non-polluted weather were collected on the summit of Mt. Tai in North China Plain. To understand the bacterial community structure and function, the Miseq 16S rRNA gene sequencing was performed, and PICRUSt predictive function was applied to examine the metabolic and ecological function. Analysis of similarities (ANOSIM) and linear discriminant analysis effect size (LEfSe) were executed to clarify the discrepant

bacterial taxa. Moreover, RDA analysis was applied to identify the pivotal environmental factor influencing bacterial community. Air mass back trajectory and wind analysis were conducted to definitude the most likely source and transmission paths of pollutants and bacteria.

## 2. Material and methods

### 2.1 Sample collection

Cloud samples were collected using the Caltech Active Strand Cloud water Collector (CASCC) on the summit of Mt. Tai ($36^o18'$ N, $117^o13'$ E, and 1534 m a.s.l). The cloud collector was cleaned prior to each cloud event and kept closed prior to cloud interception to ensure not to be contaminated. The collector was activated by a sensor only when cloud formed in the ambient air. The cloud water was aspirated through a Teflon duct at a rate of 24.5 $m^3$ $min^{-1}$ by a fan situated at the rear of collector. Cloud water collected from the Teflon strands, through Teflon tube and down into Teflon bottles. The theoretical 50% cut-off size was equivalently drop diameter of 3.5 μm.

To avoid artificial and instrumental contamination, the Teflon tube and the polyethylene bottles were first pretreated with anhydrous ethanol and then washed 3 times using the sterilized ultrapure water. Before sampling, the cloud collector was washed with the sterilized deionized distilled water filtered through the with 0.22 μm membrane. Then spray the sterilized dd-$H_2O$ into the collector using the sprayer and the collected water sample was as the blank.

To distinguish the polluted and non-polluted cloud episodes, we firstly checked the air pollution condition according to the 24 h concentration of WHO air quality guideline (25 μg/$m^3$) and this standard has been applied by Australia, New Zealand and European Union. During a cloud episode, the average $PM_{2.5}$ concentration higher than 25 μg/$m^3$ was classified as air pollution. Further definition of cloud water was combined with the major ions in water droplets, which provide deep insight into pollution levels. Therefore, in the present study, cloud episodes under high atmospheric $PM_{2.5}$ concentration and high concentration of ions in cloud water were considered as polluted.

After adjustment, seven cloud episodes including thirteen samples were detected over the whole sampling period (from 24 July to 23 August 2014), including 11 polluted and 2 non-polluted cloud water samples (Figure S1). The samples for microbial

community investigation were stored with dry ice in transit and then frozen at -80$^{o}$C in laboratory until further analysis.

In cloud water, the pH and conductivity was detected with a Multi pH/COND/TEMP 6350 hand held Meter immediately after sampling. The major inorganic ions (Cl$^-$, NO$_3^-$, SO$_4^{2-}$, Na$^+$, K$^+$, Ca$^{2+}$, Mg$^{2+}$, and NH$_4^+$) in cloud water were quantified using the ion-chromatography system (Dionex ICS-90). Hourly data, e.g., meteorological parameters, and PM$_{2.5}$ were measured to evaluate the air quality during cloud episodes (Table 1). The meteorological parameters including atmospheric visibility, temperature, relative humidity, wind direction, wind speed were measured with an automatic meteorological station (PC-4, JZYG, China) *in situ*. The mass concentration of PM$_{2.5}$ was measured using a Model 5030 SHARP monitor (SHARP 5030, Thermo Fisher Scientific, Massachusetts, USA).

To determine the most likely source region for air mass of cloud episodes, the 24-h back trajectory analysis was performed using the Hybrid Single-Particle Lagrangian Integrated Trajectories (HYSPLIT) model (http://ready.arl.noaa.gov/HYSPLIT.php). Moreover, the wind rose diagram of study area (origin, version 9.0, Origin Lab Corporation, Northampton, MA) during cloud process were analyzed to clarify the predominant wind direction and wind speed.

## 2.2 DNA Extraction and PCR Amplification

Genomic DNA was extracted in triplicate with the FastDNA spin kit for soil (MP Biomedicals, Solon, OH, USA) according to the manufacturer's directions. The concentration of DNA was determined spectrophotometrically (Nano-Drop 2000, Thermo, Wilmington, Delaware, USA). To check sample contamination, DNA was extracted through an identical extraction procedure for the blank samples from the collected dd-H$_2$O. These blanks were PCR amplified together with the DNA samples extracted from the cloud water samples. For the blank, no obvious bands and target fragment was detected by examination of electrophoretic gel images.

The designed primer sets with the V3-V4 region of 16S rRNA gene (338F-806R) (Masoud et al., 2011), adapter and barcodes were selected in the illumina Miseq sequencing. For each sample, a 25-µL PCR mix was prepared containing 10 µL of 5x Buffer, 1µL of dNTP (10mM), 1 U Phusion High-Fidelity DNA polymerase, 20 ng of template DNA, 1 µL of each 10 µM modified primer, with double-distilled water until

25 μL. PCR was performed at 94°C for 2 min; 25 cycles of 94 ℃ for 30 s, 56 ℃ for 30 s and 72 ℃ for 30 s; 72 ℃ for 5 min; and hold at 10 ℃.

The PCR products were separated by 2% agarose gel electrophoresis and purified with the nucleic acid purification Kit (AxyPrepDNA, Axygen, USA). Purified PCR products were quantified using a Qubit 3.0 fluorometer (Invitrogen, Carlsbad, CA) and then mixed to equal concentration. For each sample, 4 μL of 10 nM pooled DNA was denatured with 1 μL of 0.2 N NaOH at room temperature. Finally, the Illumina paired-end sequencing was performed on a MiSeq platform (Illumina, Inc., San Diego, CA). After sequencing, two FASTQ files (read 1 and read 2) for each sample were generated on the MiSeq reporter software automatically. Raw 16S rRNA gene sequences are available at the Sequence Read Archive (SRA) under accession number SRX1904235.

**2.3 Illumina high-throughput sequencing and analyzing**

Raw sequences were processed using the QIIME packages (Kuczynski et al., 2011). The pair-end reads were firstly merged with overlap length greater than 10 bp. Then, the adapter, barcodes and primers were removed from the merged sequences. Subsequently, the trimmed sequences with length shorter than 200 bp, quality score lower than 25, homologous longer than 8 bp, containing ambiguous characters were screened. Finally, chimeric sequences were distinguished using the Usearch61 algorithm and removed from the dataset. Optimized sequences were clustered into OTUs at the threshold of 97% similarity with the usearch61 algorithm. Single OTUs were removed and taxonomy was assigned to each OTU using the Ribosomal Database Project (RDP) classifier in QIIME, with a minimum confidence cutoff of 80% against the Silva reference database (silva 119, http://www.arb-silva.de/) to the genus level. Subsequently, we focused on the bacterial genera including species known or suspected to be opportunistic pathogen and performed a systematic literature review to identify potential pathogenic bacteria in water habitats (Bibby et al., 2010; Guo & Zhang, 2012b; Luo & Angelidaki, 2014).

To acquire bacterial community function, Phylogenetic Investigation of Communities by Reconstruction of Unobserved States (PICRUSt) was performed. The PICRUSt can be used to predict the metabolic function pathway from corresponding bacteria and archaea and provide a community's functional capabilities based on the 16S rRNA

gene sequence (Langille et al., 2013). PICRUSt has been used in bacterial diversity and function analysis (Corrigan et al., 2015; Wu et al., 2016). In the present, the phylogenetic and functional capacities for the bacteria in cloud water are of great interest to help understanding their roles in atmosphere, ecosystem and health. Bacterial community functional profiles were predicted from 16S rRNA gene using the PICRUSt program and annotated against with the Kyoto Encyclopedia of Gene and Genomes (KEGG) database. Spearman's correlation coefficients were calculated to link the pairwise comparison of KEGG pathway and genus. Selected KEGG pathways relating to metabolism and disease infection and predominant genera are included in the heatmap. Correlation is significant at P-value of less than 0.05 and 0.01.

Alpha diversity was assessed by examining the rarefaction curves, shannon-wiener curves and rank-abundance curves calculated with Mothur (v.1.34.0; http://www.mothur.org) (Schloss et al., 2009) and visualized in R project (v.3.1.3; https://www.r-project.org/). Community richness estimators including the observed OTUs, nonparametric Chao1, ACE, and community diversity estimators including Shannon and Simpson indexes were also calculated with Mothur. Moreover, the Good's coverage was used to evaluate the sequencing depth.

Principal component analysis (PCA) was performed to visualize the changes in bacterial community for the collected samples. The PCA plots were constructed depending on Bray-Curtis similarity index calculated with the abundance of OTUs using the Biodiversity package (Kindt & Coe, 2005) in R. The difference in OTU composition for samples collected in polluted and non-polluted cloud episodes was tested by the analysis of similarity (ANOSIM) (Clarke, 1993). ANOSIM was implemented with the VEGAN package in R. Linear discriminant analysis effect size (LEfSe, http://www.huttenhower.sph.harvard.edu/galaxy/) was applied to identify differentially abundant bacterial taxa associated with the polluted and non-polluted cloud episodes at genus or higher taxonomy levels (Segata et al., 2011). For all statistical tests, the P value less than 0.05 was considered significant.

## 2.4 Interaction between bacterial community structure and environmental variables

Correlation between bacterial community and environmental variables was first

performed using a detrended correspondence analysis (DCA) to estimate the length of the gradient. The resulting length (0.99) indicates a linear model was appropriate, hence RDA was subsequently performed. RDA was elaborated with the predominant bacteria data matrix and the environmental data matrix including $PM_{2.5}$ mass concentration, meteorological conditions, water pH, electric conductivity and major ions in cloud water (Anderson & Willis, 2008). Interset correlations were used to determine the most important environmental variables in determining the community structure. To explain the species data, cumulative fit per species as fraction of variance of species was analyzed. The importance for the ordination space and the species most associated with environmental factors were selected. Analysis was performed in Canonical Community Ordination for windows (Canoco, v 4.5).

## 3. Results and discussions

### 3.1 Definition of polluted and non-polluted cloud episodes

To distinguish the polluted and non-polluted cloud episodes, we firstly checked the air pollution based on the $PM_{2.5}$ concentration. In the present study, the average $PM_{2.5}$ concentration during a cloud episode higher than 25 μg/m$^3$ was classified as air pollution. We first defined cloud water samples according to the air pollution conditions. Cloud water collected under air pollution condition was considered as polluted cloud episodes. However, in PCoA and Hcluster analysis (Figure S3), sample CE1-2 considered as non-polluted was separated from the other non-polluted samples and closed to the polluted samples. Reclassification of cloud water samples were combined with the major ions in water droplets. By checking the major ions, we observed that although the $PM_{2.5}$ concentration for sample CE1-2 was low, a relative high major ions concentration was detected (Figure S1). Therefore, we categorized the sample CE1-2 as polluted sample. The cluster and PCA analysis also confirmed the classification (Figure S4).

Although the predominant bacteria are similar between polluted and non-polluted cloud episodes, significant disparity within bacterial taxa are also identified. ANOSIM analysis suggest that the OTUs from polluted samples were grouped into one large cluster, and separated from the non-polluted clusters (ANOSIM comparison, R =0.683, P=0.012, ＜0.05). Cluster analysis including PCoA and Hcluster indicated a highly similar community composition in polluted samples, regardless of the cloud

episodes (Figure S3). Cluster analysis based on the relative abundance of genera showed similar clustering patterns (Figure S4), and the polluted samples also shared high similarity in their bacterial community structure.

### 3.2 Microbial community in cloud water

Information on bacterial community in fog/cloud droplets are scarce, our study provided comprehensive investigation of bacterial community. From the 13 samples collected during 7 cloud episodes, a total of 232148 high quality sequences were obtained after quality filtering and OTUs ranged from 975 to 1258 (Table 2). This value was similar with the previous sequence-based survey of atmospheric bacteria in

dust storm (OTUs, 1214) (Katra et al., 2014), and bacteria in rain water in July (OTUs, 1542) (Cho & Jang, 2014). Identification of OTUs at different taxonomic levels yielded 359 species, 411 genera, 152 families, 70 orders, 38 classes and 26 phyla. Across all samples, Proteobacteria was the dominant phylum, followed by Bacteroidetes, Cyanobacteria, Firmicutes, Deinococcus-Thermus, Actinobacteria and

Nitrospirae (Figure 1). Bacterial community structure is similar to few other studies explored bacterial diversity in fog/cloud samples, the aforementioned phyla contained a series of genera participate in the atmospheric hydrological cycle (Amato et al., 2007b; Delort et al., 2010). They are predominant taxa in clouds at a high elevation determined by sanger sequencing and tagged pyrosequencing (Bowers et al., 2009),

and which are also the typical culturable heterotrophic bacteria from clouds broadly distributed in aquatic and terrestrial habitats (Amato et al., 2005; Kourtev et al., 2011). In the present study, Figure S5 shows the dominant genera collected during cloud process. The predominant genera from Proteobacteria (including *Acinetobacter*, *Stenotrophomonas*, *Pseudomonas*, *Sphingomonas*, *Massilia*, *Delftia*, *Brevundimonas*),

Firmicutes (*Bacillus*) and Bacteroidetes (*Empedobacter*) were similar across all samples. The identified genera in cloud water were similar to the limited data described microorganisms in fog/cloud water. Fuzzi et al. (1997) investigated bacteria in fog droplets in a highly polluted area and found the predominant genera from *Pseudomonas*, *Bacillus* and *Acinetobacter*. Amato et al (2007b) observed more

diverse genera from the phylum of Proteobacteria, Bacteroidetes, Actinobacteria and Firmicutes, which mainly belonging to *Pseudomonas*, *Sphingomonas*, *Staphylococcus*, *Streptomyces* and *Arthrobacter*. Ahern et al (2006) investigated bacterial community

in clouds collected in Scotland and found the dominant species were from *Pseudomonas* and *Acinetobacter*.

Bacterial community function was estimated with PICRUSt algorithm. After PICRUSt analysis, pathways with participants less than 10% were removed, leaving 225 non-human-gene KEGG pathways. These predominant pathways showed in Figure S6 were mainly related to Amino Acid Metabolism, Carbohydrate Metabolism, Cell Motility, Cellular Processes and Signaling, Energy Metabolism, Enzyme Families, Folding, Sorting and Degradation, Membrane Transport, Nucleotide Metabolism, Nucleotide Metabolism, Replication and Repair, Signal Transduction, Transcription, Translation. Besides the pathways associated with microbial physiological metabolism, we focused on the microbial pathways of metabolic processes in a variety of environments. Fog/cloud droplets contains carbon and nitrogen compounds, which could be available substrate for microbial metabolism in the atmosphere. The predicted function of metabolism was likely attributed to the bacterial gene from the identified taxa (Figure 2). Previous studies have demonstrated that atmospheric bacterial community contained a metabolically diverse group found in a wide range of water/soil habitats. For example, *Acinetobacter*, the most abundant genera widely distributed in land or ocean, was positively associated with the biodegradation, leaching and removal of several organic and inorganic man-made hazardous wastes (Abdelelhaleem, 2003). *Stenotrophomonas* and *Pseudomonas*, positively correlation with carbohydrate metabolism and glycan biosynthesis and metabolism, are well-known for the striking capability to utilize numerous carbon sources, have been widely utilized in the degradation and transformation of complex organic compounds in a wide range of habitats (Boonchan et al., 1998; Stanier et al., 2010). Moreover, predicted functions associated with human disease are especially concerned. For instance, some species from *Acinetobacter*, were positively associated with infection disease (Nemec et al., 2001). *Empedobacter* from Bacteroidetes widely distributed in water habitats, are human clinical origins, certain species from *Empedobacter* are ranked as potential pathogens (Hugo et al., 2005).

In cloud water, a variety of genera adapt to harsh environments were also identified. *Sphingomonas*, the ability to survive in low concentrations of nutrients has been reported, which can metabolize a series of carbon compounds, events toxic compounds (Xu et al., 2006). Similar to *Sphingomonas*, members of *Brevundimonas* are well known for their ability to withstand extreme harsh environment (Kopcakova

et al., 2014). The spore forming bacteria *Bacillus* included in the phylum Firmicutes are commonly airborne bacteria found in bioaerosol, cloud water, rainwater and could survive in cold environment (Després et al., 2012). Similar to *Bacillus*, some strains of *Pseudomonads* found in Antarctic environments revealed the cold adaption (Bozal et al., 2003). Certain *Pseudomonads* species found in cloud water were psychrophiles, they grow faster at 5 ℃ than at high temperature (17 ℃ or 27 ℃) (Amato et al., 2007b). Members of *Deinococcus* from Deinococcus-Thermus are available to withstand extreme radiation conditions that could potentially adapt to the cloud environment (Mattimore & Battista, 1996).

Although most the bacterial ecophysiological role in biogeochemical cycles is generally established based on soils and water habitats, information about bacterial activity in cloud water is available. The identification of microorganisms under barren nutrition, low temperatures and radiation environment encountered in clouds is expected since similar bacterial species have been retrieved and proved to be active in harsh environments. Their adaption to the specific environments in fog/cloud water with the potential role in the nucleation, metabolism of organic pollutants, demonstrated the potential importance in participation and influence atmospheric biochemistry cycle.

## 3.3 Implications in human health and ecosystem

Bacteria in fog/cloud water have been discovered for decades but detailed information on community composition and potential ecophysiological role is severely limited. Bioaerosols in fog/cloud have been complex assemblages of airborne and exogenic microorganisms, likely emissions and resuspension from various terrestrial environments, e.g., soil, water, plants, animals or human beings. In the atmosphere, fog/clouds may be favorable niche for bacteria and these bacteria could thrive and influence cloud processes by acting as cloud condensation nuclei and ice nuclei. Bacteria including pathogenic or beneficial species can also be attached to particles or incorporated into water droplets of fog/clouds. During fog/cloud or rain process, they can be deposited back to land via deposition and possibly induce infections to human health and impose effect on the diversity and function of aquatic and terrestrial ecosystems (Kaushik & Balasubramanian, 2012; Simmons et al., 2001; Vaïtilingom et al., 2012) (Figure 3 and Table 3).

Atmospheric bacteria are efficient cloud condensation nuclei and water vapour can be condensed on bacterial cell surface (Mohler et al., 2008). The hygroscopic growth of bacteria below water saturation and supersaturations has been observed for some species, e.g. Bauer et al. (2003) found that *Brevundimonas diminuta* was activated at $<$ 0.1% supersaturation (Bauer et al., 2003). In addition, various strains of *Pseudomonas*, *Rhodococcus* and *Bacillus* found in cloud water samples could produce biosurfactants and act as cloud condensation nuclei (Delort et al., 2010). They may form cloud droplets combined with aerosol particles at lower supersaturations and quickly grow to large size droplets and facilitate rain formation (Mohler et al., 2007). Moreover, *Pseudomonas* could induce ice nucleation at a warmer temperature than usual (Amato et al., 2015). Simulations experiments about cloud forming suggest that *Pseudomonas* was first acted as CCN, then induced freezing and ice nucleation process (Mohler et al., 2008). In addition to *Pseudomonas*, other bacteria from *Acinetobacter*, *Bacillus*, *Flavobacterium*, *Sphingomonas*, and *Stenotrophomonas* sp. (Table 3), were ice nucleation active (Mortazavi et al., 2008). Gaining an understanding of possible role in cloud condensation and ice nucleation processes might open a new sight of bacterial communities influence on meteorology and climate change.

In addition to the bacteria survive in cold environments and act as efficient cloud condensation nuclei or ice nuclei, the presence of microorganisms in fog/cloud may play vital role in atmospheric biochemistry. The detection of bacteria in cloud water associated with biotransformation of organic compounds raised a general understanding of the potential role in atmospheric chemistry. The identified species from the genera of *Rhodococcus*, *Sphingomonas*, *Delftia*, *Comamonas* (Table 3) were mainly participated in the biodegradation of organic compounds. Excessive studies have illustrated their capability of metabolism of hydrocarbon compounds, even toxic pollutants, e.g., aromatic compounds (Bock et al., 1996; Busse et al., 2003; Geng et al., 2009; Goyal & Zylstra, 1996). Two strains from *Stenotrophomonas* (*S. rhizophila*) and *Phyllobacterium* (*P. myrsinacearum*) are typical rhizospheric microorganisms, which were typically dispersed into atmosphere from soil. As plant-associated strains, *S. rhizophila* fulfill plant-protective roles and have been safely applied in biotechnology (Alavi et al., 2013). *P. myrsinacearum* is a predominant rhizospheric bacterium, which has been utilized in plant growth promotion and biological control

of soil-borne diseases due to its capability of azotification (Gonzalezbashan et al., 2000). The methylotrophic bacteria *Methylobacterium* (*M. aquaticum* and *M. adhaesivum*) are typically inhabit in soil and water. Previous studies have demonstrated the carbon fixing function in ecosystem (Gallego et al., 2006; Gallego et al., 2005). Similar to *Methylobacterium*, *Cyanobacterium* sp., widely distributed in soil, water, and various arid environments, have excellent nitrogen and carbon fixing ability (Jha et al., 2004). Cloud water seems to harbor highly diverse bacterial communities in ecosystem, which may be the atmospheric mixing of diverse point sources origins in the rhizosphere, soil and water and possibly participate in the biodegradation of organic compounds in cloud water.

In addition, after sequencing, bacterial genera containing potential pathogens were especially concerned. By blast with the reference pathogen database, sequences high similar with potential pathogens were identified. In the present study, the presence of potential pathogen sequences indicated occasional distribution and dispersion of pathogens in cloud water (Table 3). The identified opportunistic pathogens from *Empedobacter*, e.g., *E. brevis*, can be easily isolated from clinical resources, which may be associated with eye infections (Bottone et al., 1992). Occurrence of *Staphylococcus equorum* in cloud water can be expected since *Staphylococcus* are frequently isolated from airborne samples (Seo et al., 2008). They can reside on the skin and mucous membranes of humans and induce severe infections (Nováková et al., 2006). Similarly, species from *Brevundimonas* (*B. vesicularis* and *B. diminuta*) can induce virulent infections, often associated with nervous system or bacteraemia (Gilad et al., 2009; Han & Andrade, 2005). Besides that, the pathogenic strains from *Acinetobacter* (*A. schindleri*) and *Moraxella* (*M. osloensis*) are associated with skin and wound infections, bacteremia and pneumonia (Banks et al., 2007; Nemec et al., 2001).

Previous studies on potential pathogens are mostly focused on the atmospheric particulate matter ($PM_{2.5}$ and $PM_{10}$) (Cao et al., 2014; Creamean et al., 2013), rain water (Kaushik & Balasubramanian, 2012; Simmons et al., 2001), and indicated that health risk-related bacteria in atmospheric samples should be concerned. For cloud/fog water, studies of health risks to individuals are typically focused on the chemical characteristic, e.g., the low pH (acid fog) (Hackney et al., 1989), PAH (Ehrenhauser et al., 2012), etc. Limited literature discussed the microorganism in fog/cloud water suggested potential pathogens in fog/cloud water (Vaïtilingom et al.,

2012), they find potential plant pathogens such as *Pseudomonas syringae* and *Xanthomonas campestris* and suggest these living plant pathogens could then infect new hosts through precipitation. Possibly, greater survival of human pathogens may be supported in the atmosphere. Fog/cloud and rain process are part of atmospheric life cycle and dispersal pathway for some pathogenic bacteria. Studies of the airborne dispersal of pathogenic bacteria, e.g., *Neisseria meningitides*, *Staphylococcus aureus* from dust samples from Kuwait and *Pseudomonas aeruginosa* in USA Virgin Islands have indicated the spread of specific human and plant diseases over long term transport in atmosphere (Griffin, 2007; Griffin et al., 2003; Griffin et al., 2006). However, detailed health risk-oriented studies induced by pathogenic microorganisms should be deeply conducted and prudently assessed. Further study depending on the culture-dependent method and biochemical experiments will perform to check the pathogenicity.

**3.4 Disparity between polluted and non-polluted cloud episodes**

To distinguish indicator species within the polluted and non-polluted cloud episodes, LEfSe is performed, which showed statistically significant differences. A total of 70 bacterial groups were distinct using the default logarithmic (LDA) value of 2. Cladograms show taxa with LDA values higher than 3.5 for clarity (Figure 4). Consequently, 8 and 19 represent bacterial taxa in polluted and non-polluted cloud episodes were detected.

In polluted cloud episodes, most enriched bacteria were ranked as opportunistic pathogens, such as Proteobacteria, Gammaproteobacteria, Xanthomonadales, Xanthomonadaceae, Stenotrophomonas, Moraxellaceae and *Acinetobacter*. Like Proteobacteria, the Gram-negative Gammaproteobacteria contains a series of ecologically and medically important bacteria, e.g., the pathogenic Enterobacteriaceae, Vibrionaceae and Pseudomonadaceae. The Xanthomonadales from this phylum has been reported to cause disease in plants (Saddler & Bradbury, 2005). Certain species from *Stenotrophomonas* (Gammaproteobacteria, Xanthomonadales) are associated with multiple human infections. *Moraxella* form Moraxellaceae (Gammaproteobacteria) has been reported to associate with septic arthritis of the ankle (Banks et al., 2007). As previous mentioned, species from the genus of *Acinetobacter* are opportunistic pathogens and cause severe clinical infections

(Nemec et al., 2001).

In comparison, the majority of indicator species in the non-polluted samples are from Bacteroidetes, Firmicutes, Betaproteobacteria and Deinococcus-Thermus. An important biomarker from Bacteroidetes is Flavobacteriia, relative study has illustrated the marine sources for Flavobacteria. Most of Flavobacteria sequences searched by NCBI blast are mainly from marine sources, i.e., algae, oysters and sea cucumbers (Cho & Jang, 2014). The genera *Empedobacter* was abundant across all samples, which are included in the family Flavobacteriaceae. As mentioned above, *Empedobacter* (Firmicutes) are potential pathogens and resistant to a wide range of antimicrobials (Hugo et al., 2005). Clostridiales (Clostridia) and *Bacillus* (Bacillaceae) are two represent biomarkers from Firmicutes identified in the non-polluted cloud water samples. As ubiquitous in nature, these two groups contain some medically significant species (Miller et al., 2001; Makino & Cheun, 2003). Moreover, their specific physiological characteristics (produce a variety of enzymes and metabolites) and excellent ability to decomposition of organic matter have made the widely utilization in biotechnology and fermentation industry (Doi et al., 1992; Łoś, et al., 2010). Members of Oxalobacteraceae (Betaproteobacteria) are rhizosphere microorganisms involved in the biological nitrogen fixation (Donn et al., 2015). The Burkholderiales (Betaproteobacteria) commonly found in water and soil are involved in the biodegradation of various aromatic compounds (Pérez-Pantoja et al., 2012). Deinococci, from the phylum of Deinococcus-Thermus could resistant to extreme radiation and survive in extremes of heat and cold (Griffiths & Gupta, 2007).

By comparison, potential pathogens were significant groups in the polluted samples, whereas a diverse ecological function group was identified in the non-polluted samples originated from a wide range of habitat. Ecologically meaningful distinguish of bacterial groups under polluted and non-polluted conditions is essential for understanding the structure and function, and which provide a general understanding of bacterial metabolism in cloud water.

## 3.5 Environmental factors shaping bacterial community structure

To clarity the vital environmental factor in shaping bacterial community structure, RDA was performed to discern the genus-level structure with the selected environmental factors (Figure 5). The first two axes explained 65.9% of the

accumulated variance in the species-environment relation. Interset correlations showed major ions and $PM_{2.5}$ was the most important environmental variables structuring the bacterial community (axis 1, major, -0.436; $PM_{2.5}$, -0.367); in turn, wind speed and temperature registered the high value for axis 2 (wind speed, -0.509; temperature, -0.494) (Table S1).

Cumulative fit indicated that the predominant genera affiliated with *Acinetobacter*, *Bacillus*, *Corynebacterium*, *Phyllobacterium*, *Pseudoalteromonas* and *Rhodococcus* displayed strong correlations with axis 1. *Empedobacter*, *Hydrotalea*, *Paracoccus*, *Pelomonas*, *Pseudomonas* and *Stenotrophomonas* were notable genera highly correlations with axis 2 (Table S2). As aforementioned in section 3.3, the above bacteria were metabolically diverse groups found in various habitats and certain genera included potential pathogens.

Of the environmental characteristics measured, major ions in cloud water and atmospheric $PM_{2.5}$ were the best predictors of diversity variability of bacterial community structure. These two parameters were strongly correlated with represent bacterial genera. As indicated in Figure S1, significant positive correlation was observed between major ions (x) and $PM_{2.5}$ (y), $y=0.00477x+5.324$, $p<0.01$, $R^2=0.757$. Relevant studies suggest that bacterial community was highly variable under $PM_{2.5}$ mass concentration (Cao et al., 2014). Statistical analysis, e.g., correlation or multiple linear regression, indicated that $PM_{2.5}$ exhibited a negative correlation with airborne bacteria in haze days (Gandolfi et al., 2015; Gao et al., 2015), whereas in another study, spearman correlation analysis showed $PM_{2.5}$ exhibited a significant positive correlations with the airborne microbe concentration (Dong et al., 2016). Possibly, the inorganic and organic compounds in particulate matter ($PM_{2.5}$) can be available nutrients for microbial growth in air. However, aggregation of harmful substances such as heavy metals, polycyclic aromatic hydrocarbons would be toxic for bacteria under high $PM_{2.5}$ mass concentration.

During cloud process, most atmospheric particles (including $PM_{2.5}$) are scavenged in cloud water. In polluted air, high $PM_{2.5}$ concentration resulted in the elevated water soluble inorganic ions in cloud water droplets. Therefore, similar trends were observed between major ions and $PM_{2.5}$ concentration. In cloud water, the water soluble major ions and the microorganisms co-exist in the same microenvironment. Major ions in cloud droplets could provide available nutrition for bacterial growth and

duplication. Previous study has suggested that these nutrients are related to the distribution of bacteria in water habitats (Newton et al., 2011). Meanwhile, PICRUSt analysis in section 3.2 discovered a series of metabolic pathway involved in the bacterial basic physiological activities and carbon, nitrogen, sulfur metabolism (Figure S6). For example, the sulfate, nitrate and ammonium can be available substrates for bacterial growth, magnesium and calcium are involved in a series of physiological activities (e.g., signal regulation, transmembrane transport) (Fagerbakke, et al., 1999; Fiermonte, et al., 2004; Michiels et al., 2002). Therefore, major ions were important environmental factor shaping community structure in cloud water. $PM_{2.5}$ played an indirect role by influencing the concentration of major ions in water droplets.

The identified taxa either from polluted or non-polluted samples were typically found in soil, water, plant or human beings. These bacterial groups aerosolized and dispersed into atmosphere either from local regional emissions or long-term transport. Source tracking analysis by backward trajectory indicated that air mass of polluted cloud episodes came largely from northern and western China, moved east through Shanxi, Henan, Hebei province to the study area, or from outer Mongolia, crossing the Jingjinji area to Mt.Tai (Figure 6). The passed areas were notable as heavy industry region with frequent coal mining activities and serious pollution. Moreover, the large population and agricultural activities resulted in the numerous pathogenic microorganisms from human or animal fecal dispersed in the atmosphere. In contrary, air mass of non-polluted cloud episodes originated mostly from the southern China, and the passed region were rich of water resources, e.g., Dongting Lake, Huaihe river, Yangtze river etc. The marine sources bacteria (Flavobacteria, significant biomarker in non-polluted cloud water samples by LefSE, Figure 4) dispersed in the atmosphere typically derived from the evaporation of lakes and rivers water. These bacteria mainly originated from sea-air interactions and airborne marine bacteria can be transported to inland through long-term transport.

In the sampling site (the sumit of Mt.Tai, 1534 m a.s.l), local anthropogenic pollution might be minimized and air pollution is mainly influenced by long term transport. Wind rose diagram suggest the predominant west wind during polluted cloud episodes and wind speeds ranged 1.2-1.4 m/s, whereas in non-polluted cloud episodes it was mainly southwest wind with higher wind speed (2.4-3 m/s) (Figure 7). Wind direction and speed are important meteorological factors influencing fog/cloud formation (Fu et

al., 2014). Recent studies also indicate the variation of bacterial concentration and community structure conducted by wind (Evans et al., 2006; Jones & Harrison, 2004). In addition, wind and $PM_{2.5}$ distribution graph (Figure 7C) indicates that during the whole sampling time (from 24 July to 23 August 2014) $PM_{2.5}$ concentration was high under lower wind speed, whereas lower $PM_{2.5}$ was observed when wind speed was high. Air mass from the contaminated area through long-term transport combined with lower wind speeds, largely reduce the diffusion rate of pollutants and thus lead to the sustained high $PM_{2.5}$ during polluted cloud episodes. In polluted air, the soluble composition in $PM_{2.5}$ was accumulated in atmosphere and could be dissolved in cloud water droplets during wet deposition. Therefore, the concentration of water soluble ions increased under high $PM_{2.5}$ concentration, which has directly influence on microbial community. Whereas in the non-polluted cloud episodes, higher wind speed was beneficial to the diffusion of pollutants and resulted in the lower $PM_{2.5}$ mass concentration, which finally led to the significant discrepancy of bacterial community structure. However, further research still needed to address the detailed interaction between bacterial community and environmental factors, and to understand the mechanism that how chemical composition influence microbial community.

## 4 Conclusion

The composition and potential function of microbial communities in atmospheric water phase (fog and clouds) remained rarely studied. Using the 16S rRNA gene sequencing, this work provided a thorough investigation on bacterial ecological diversity under polluted and non-polluted cloud episodes and revealed a highly diverse bacterial community harbored in cloud water. Correlation analysis for the predominant genera and PICRUSt function predication enhanced the understanding of the distribution of bacteria and their potential involvements in atmosphere, ecosystem and human health. The identification of bacteria survive in barren nutrition, low temperatures and radiation environment encountered in fog/cloud water demonstrated bacterial active in harsh atmospheric environments. They may act as efficient cloud condensation nuclei or ice nuclei, associate with biogeochemical cycling (nitrogen/carbon cycling) and microbial degradation of organic compounds in fog/clouds and spreading of specific human, animal and plant diseases by potential pathogens. Moreover, community disparity between polluted and non-polluted cloud

episodes suggested that major ions in cloud water and atmospheric PM$_{2.5}$ seem to be pivotal variables in shaping bacterial community. PM$_{2.5}$ had potential impact on bacterial community structure by influencing major ions in water droplets, which is likely to provide a more comprehensive understanding of the atmospheric microbial biodiversity under environmental stress. These results provide a basic understanding of mechanism of bacterial community response and metabolism in polluted weather for further study.

## Acknowledgement

This work was supported by National Natural Science Foundation of China (41605113, 41375126), Taishan Scholar Grand (ts20120552), China Postdoctoral Science Foundation (No. 2015M582095).

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

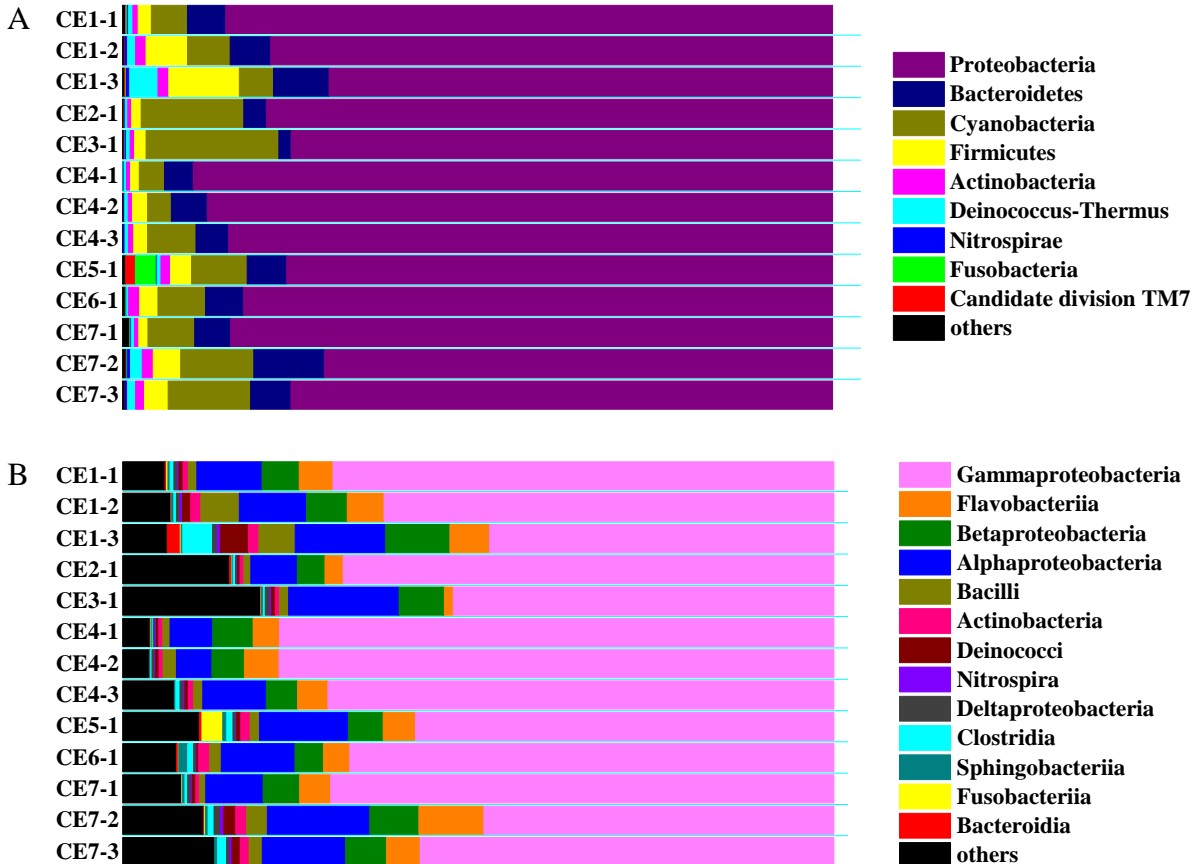

**Figure 1** Bacterial community variation for cloud episodes at the phylum (A) and class (B) level. CE refers to the cloud episodes. Bar graphs for each sample represent the percentage of taxa assigned to each phylum with 80% bootstrap confidence. Taxonomic summary of the most abundant taxa (more than 1%) across all cloud samples are indicated in the figure.

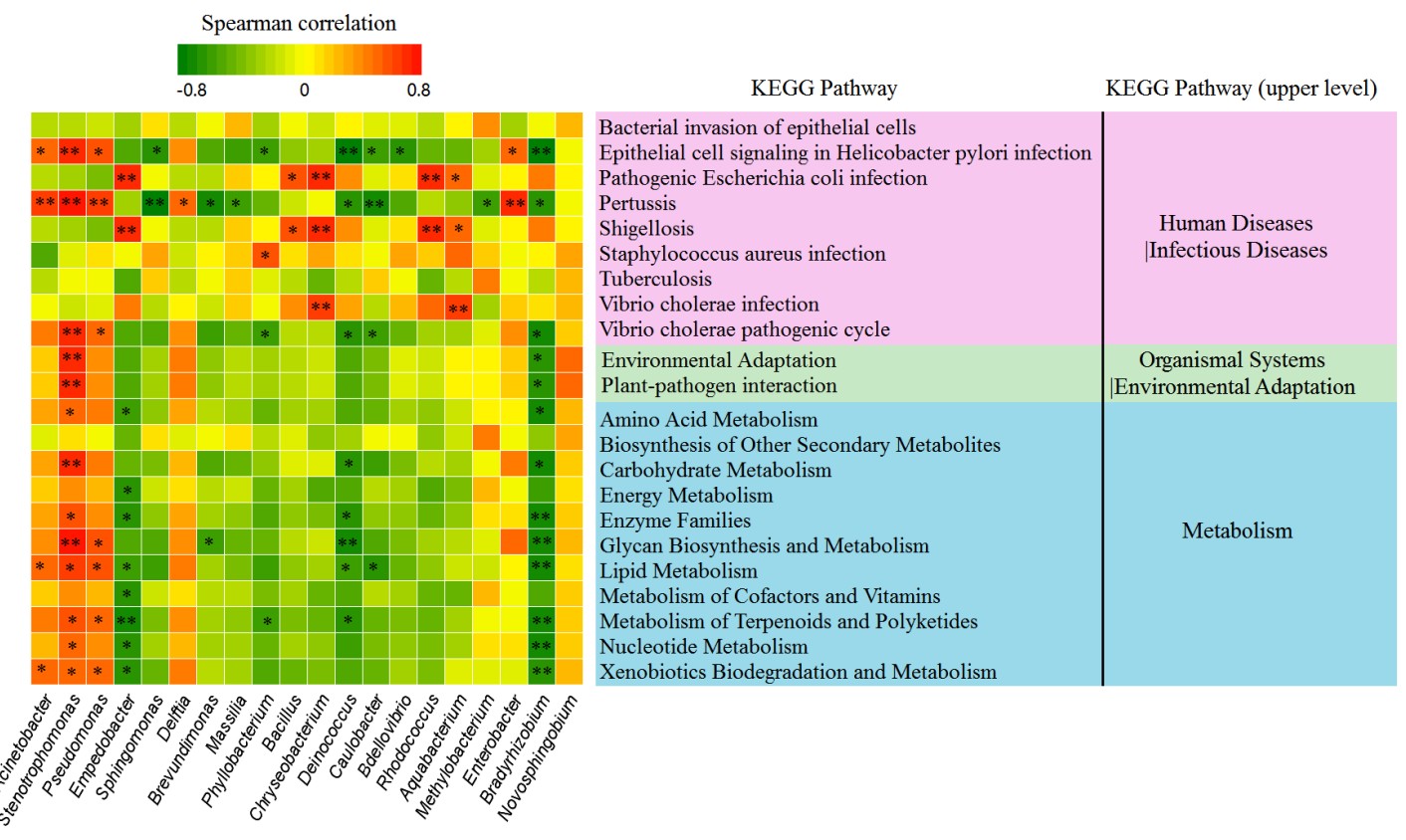

**Figure 2** Bacterial taxa are related to KEGG functional pathways. Bacterial gene functions were predicted based on 16S rRNA gene sequences using the PICRUSt algorithm and annotated from KEGG databases. Spearman's correlation coefficients were calculated for each pairwise comparison of genus and KEGG pathway. Selected KEGG pathways related to metabolism and disease infection and predominant genera are included in the heatmap. Red color refers to the positive correlation, and green indicates a negative correlation. Correlation is significant at *P < 0.05, **P < 0.01.

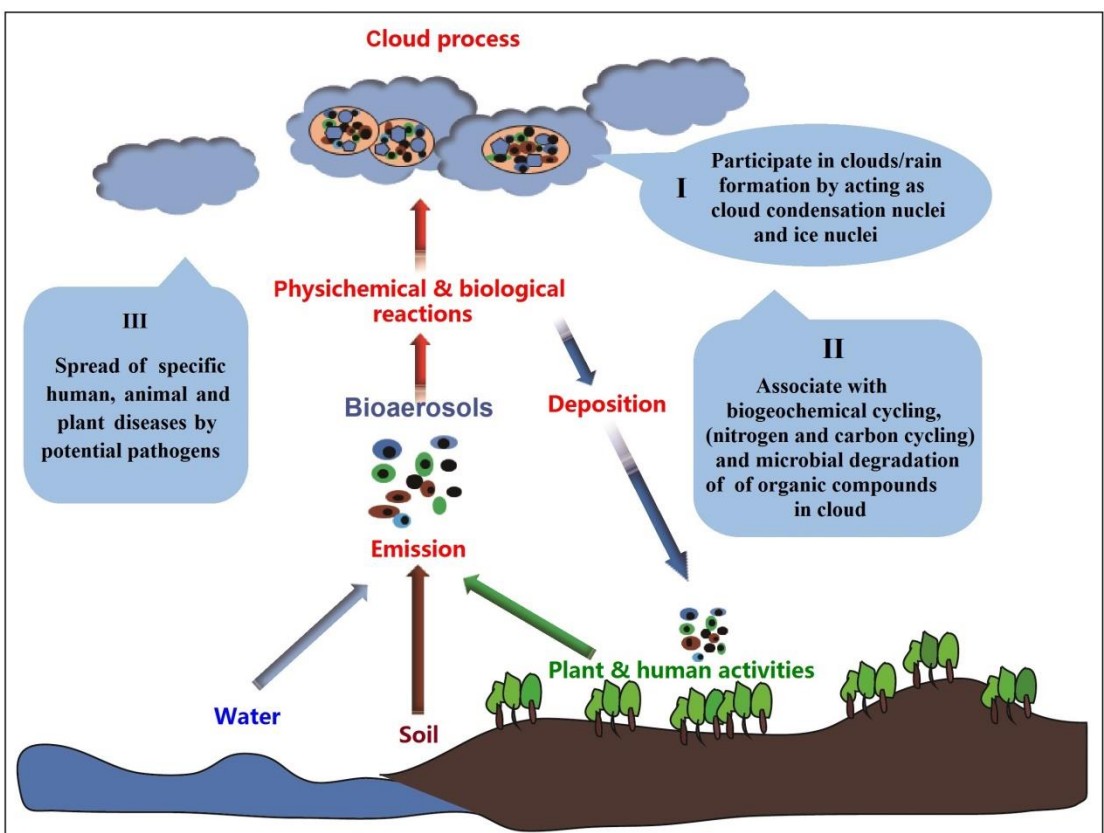

**Figure 3** Schematic representation of bioaerosols life cycle and potential influence on atmosphere, ecosystem and human health, modified from Poeschl (Poeschl, 2006). The predominant bacteria species with potential functions are indicated in the figure.

Bioaerosols emissions and resuspension from various terrestrial environments, e.g., soil, water, plants, animals or human beings, may include pathogenic or functional species. These bacteria can be attached to particles or incorporated into water droplets of clouds/fog. Certain species can serve as biogenic nuclei for Cloud Condensation Nuclei (CCN) and Ice Nuclei (IN), which induce rain formation, precipitation, and wet deposition of gases and particles. For the potential pathogens and functional bacteria, during cloud process, they can be deposited back to land via deposition and possibly induce infections to human health and impose effect on the diversity and function of aquatic and terrestrial ecosystems.

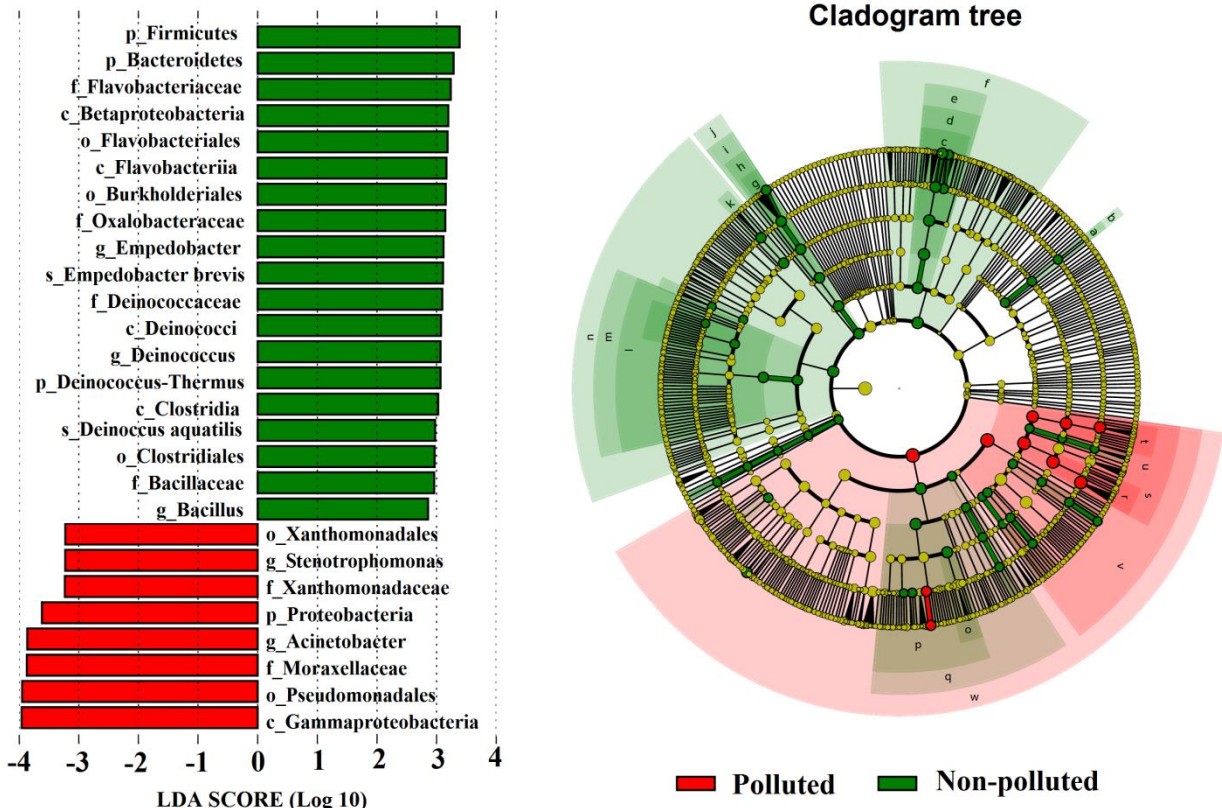

**Figure 4** Distinct bacterial taxa between polluted and non-polluted cloud episodes identified by linear discriminant analysis coupled with effect size (LEfSe). The LDA effect sizes (left) were calculated using the default parameters. The taxonomic cladogram (right) was visualized with LDA values higher than 3.5 comparing all bacterial taxa. The significantly distinct taxon nodes are colored in red and green for polluted and non-polluted cloud episodes, respectively. Bacterial taxa with nonsignificant differences are indicated with yellow circles and circle diameter is proportional to relative abundance. The abbreviation in the cladogram tree: a: g_Rhodococcus, b: f_Nocardiaceae, c: f_Flavobacteriaceae, d: o_Flavobacteriales, e: c_Flavobacteriia, f: p_Bacteroidetes, g: f_Deinococcaceae, h: o_Deinococcales, i: c_Deinococci, j: p_Deinococcus-Thermus, k: f_Bacillaceae, l: o_Clostridiales, m: c_Clostridia, n: p_Firmicutes, o: f_Oxalobacteraceae, p: o_Burkholderiales, q: c_Betaproteobacteria, r: f_Moraxellaceae, s: o_Pseudomonadales, t: f_Xanthomonadaceae, u: o_Xanthomonadales, v: c_Gammaproteobacteria, w: p_Proteobacteria.

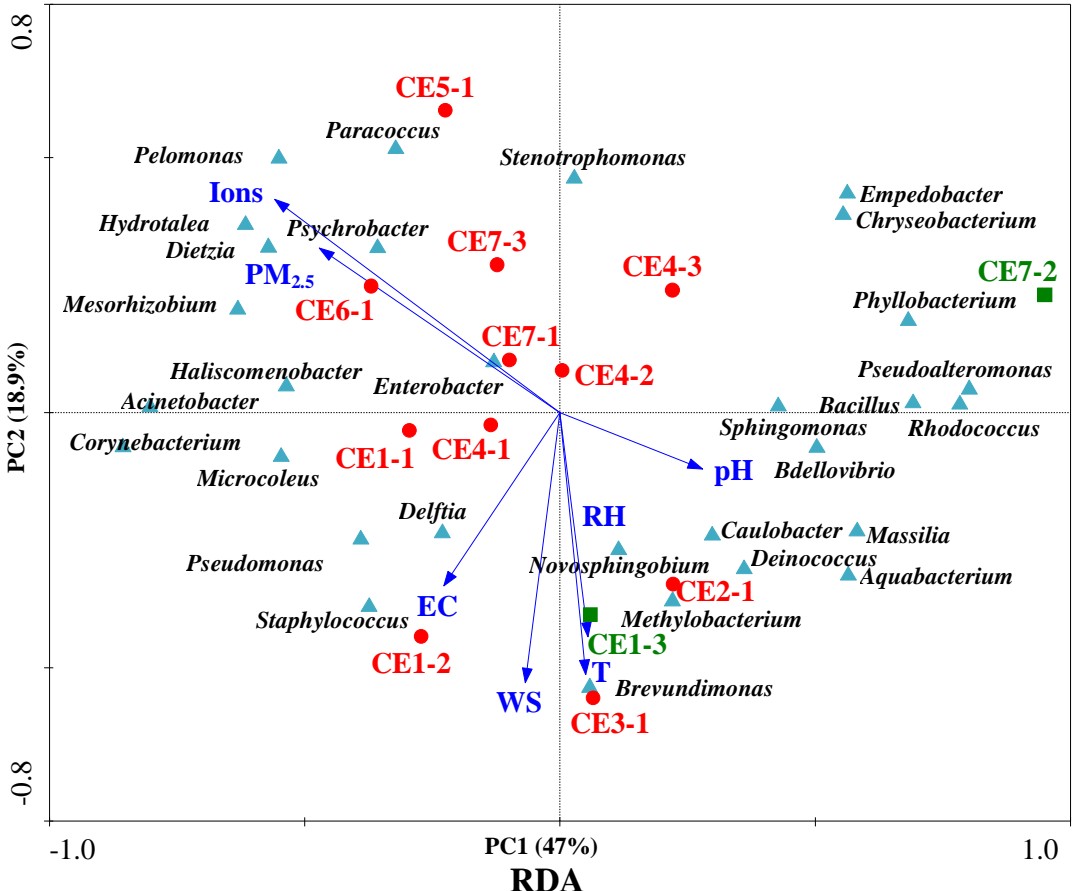

**Figure 5** Biplot of the environmental variables and genus-level community structure using a redundancy analysis (RDA) model, describing the variation in bacterial community explained by environmental variable. CE refers to cloud episodes. Polluted episodes are indicated in red circle, and non-polluted episodes are green squares. Species data are listed in Table S2. The selected environmental variables are significant ($P < 0.05$) using Monte Carlo permutation testing. Species are labeled with triangle, the closer of which indicates existence in similar environment. Environmental variables are showed by arrows; the relative length is positive correlation with the importance in influencing bacterial community structure. The angle between the arrow and the ordination axis suggest the variable response respect to the RDA gradient. The two axes explain 65.9% of the variability. For bacteria, major ions in cloud water seem to be the most important environmental variable shaping the community structure.

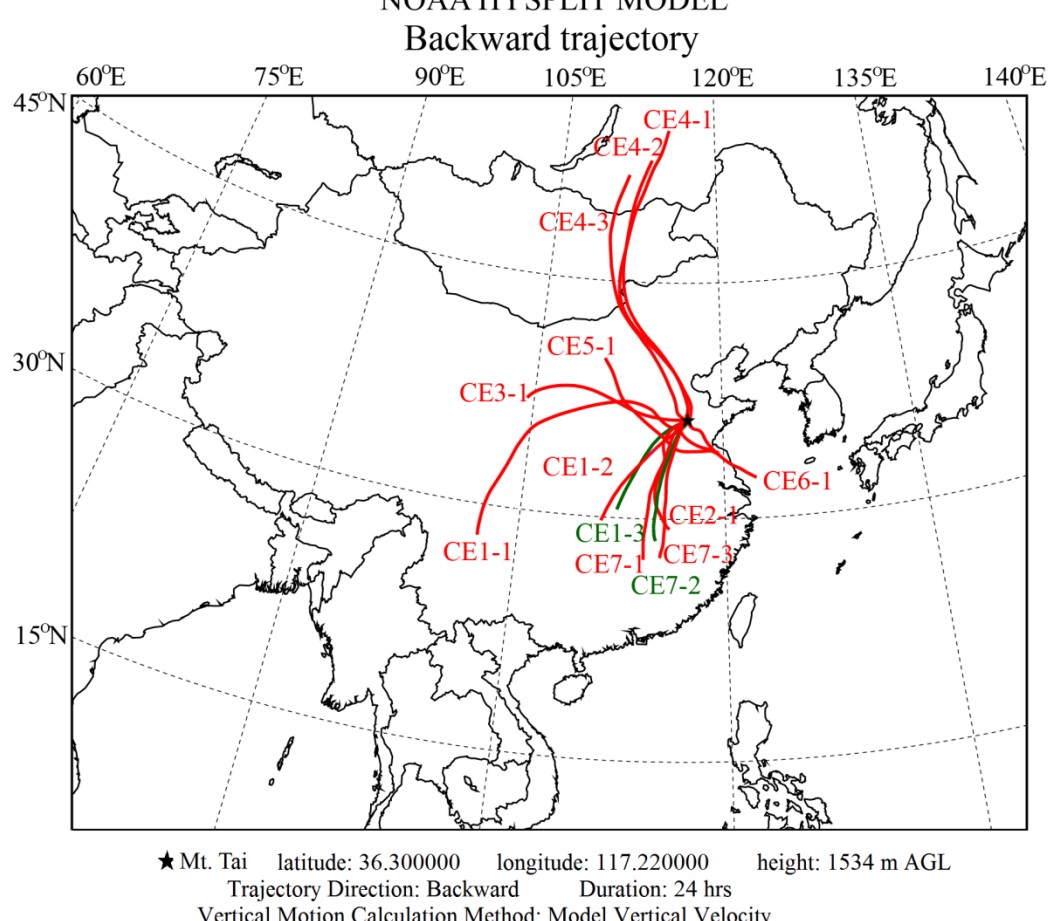

**Figure 6** Air mass transport pathways for the cloud episodes using the Hybrid Single Particle Lagrangian Integrated Trajectory (HYSPLIT) model. 24-hour backward trajectories were calculated for air parcels arriving at the summit of Mt. Tai ($36^{o}18'$ N, $117^{o}13'$ E, and 1534 m a.s.l). CE refers to cloud episodes. The polluted episodes are indicated in red lines, and green lines are non-polluted episodes.

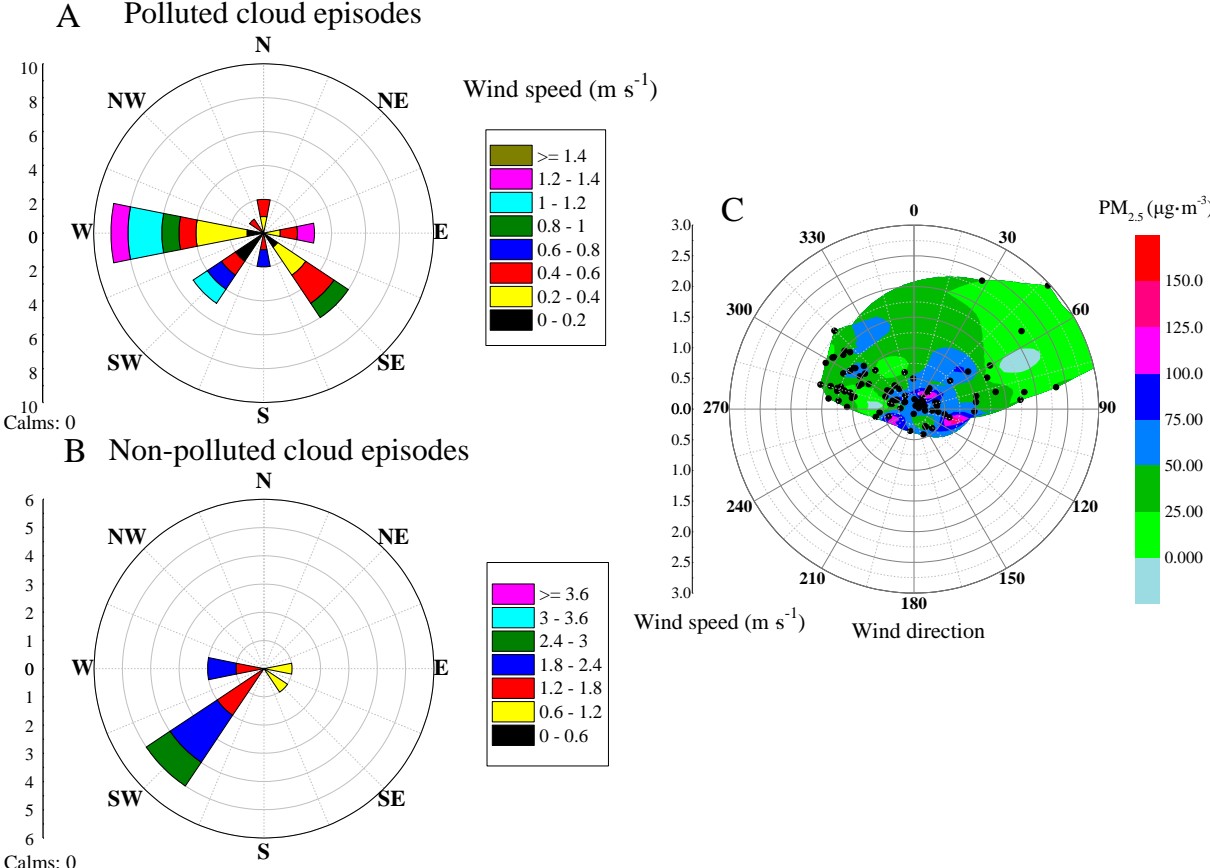

**Figure 7** Wind Rose Diagram to quantitative analysis of wind speed and wind direction during sampling time between polluted (A) and non-polluted cloud episodes (B). The frequency of winds is indicated by wind direction. Wind speed range is labeled with color bands. Wind direction with the greatest frequency is shown with the direction of the longest spoke. Figure 7 (C) indicates distribution of wind speed during the whole sampling time (from 24 July to 23 August 2014) and correlation with $PM_{2.5}$ concentration. As shown in the figure, $PM_{2.5}$ concentration was high under lower wind speed, whereas $PM_{2.5}$ was lower when wind speed was high.

## Table 1 Description cloud episodes at Mt. Tai, China

| Data | Samples | Start time (BJT) | Stop time (BJT) | Duration (h) | $PM_{2.5}$ [a] ($\mu g \cdot m^{-3}$) | LWC ($g\ m^{-3}$) | pH | EC ($\mu S \cdot cm^{-1}$) | OC ($mg\ L^{-1}$) |
|---|---|---|---|---|---|---|---|---|---|
| 24 Jul 2014 | CE1-1 | 8:50 | 15:30 | 6:40 | 105.07 | 0.21 | 4.03 | 583 | ND |
| | CE1-2 | 15:30 | 17:30 | 2:00 | 22.35 | 0.23 | 4.32 | 219.2 | ND |
| | CE1-3 | 17:30 | 22:51 | 5:21 | 14.66 | 0.24 | 5.74 | 104.4 | ND |
| 5 Aug 2014 | CE2-1 | 6:45 | 9:17 | 2:32 | 30.36 | 0.22 | 5.80 | 275.7 | ND |
| 5 Aug - 6 Aug 2014 | CE3-1 | 19:05 | 4:01 | 8:56 | 42.25 | 0.10 | 5.10 | 501 | ND |
| 14 Aug - 15 Aug 2014 | CE4-1 | 22:41 | 0:44 | 2:03 | 42.69 | 0.02 | 6.36 | 170.4 | BDL |
| | CE4-2 | 0:44 | 5:06 | 4:22 | 47.98 | 0.03 | 5.34 | 86.34 | 0.04 |
| | CE4-3 | 5:06 | 6:03 | 0:57 | 36.88 | 0.02 | 4.89 | 64.95 | BDL |
| 17 Aug 2014 | CE5-1 | 10:10 | 11:18 | 1:08 | 63.18 | 0.39 | 5.20 | 120.5 | 0.11 |
| 17 Aug - 18 Aug 2014 | CE6-1 | 22:18 | 1:25 | 3:07 | 54.33 | 0.10 | 3.80 | 321.8 | 0.02 |
| 23 Aug 2014 | CE7-1 | 2:30 | 4:38 | 2:08 | 30.45 | 0.20 | 4.38 | 356.2 | 0.03 |
| | CE7-2 | 4:38 | 6:21 | 1:43 | 23.39 | 0.22 | 5.01 | 207.5 | 0.15 |
| | CE7-3 | 6:21 | 9:20 | 2:59 | 41.60 | 0.21 | 5.74 | 187.6 | 0.21 |

CE refers to cloud episode. BJT refers to Beijing Time, which equals UTC + 8. LWC refers to the cloud liquid water content.

EC refers to the electric conductivity. OC refers to the organic carbon in cloud water.

ND means not detected due to instrument failure. BDL means below detection limitation.

**Table 2 Summary of bacterial diversity and richness of cloud water**

| Sample ID | Reads | OTUs | Ace | Chao1 | Coverage | Shannon | Simpson |
|---|---|---|---|---|---|---|---|
| Polluted cloud episodes | | | | | | | |
| CE1-1 | 18213 | 975 | 1835 | 1491 | 0.9761 | 3.9418 | 0.0646 |
| CE1-2 | 18702 | 1184 | 1841 | 1730 | 0.9719 | 4.1919 | 0.0620 |
| CE2-1 | 19914 | 1125 | 1756 | 1684 | 0.9744 | 3.9582 | 0.0630 |
| CE3-2 | 18199 | 1022 | 2082 | 1582 | 0.9734 | 3.9749 | 0.0647 |
| CE4-1 | 18350 | 941 | 1828 | 1461 | 0.9762 | 3.6041 | 0.0953 |
| CE4-2 | 17707 | 967 | 1522 | 1427 | 0.9752 | 3.6748 | 0.0902 |
| CE4-3 | 17397 | 981 | 2091 | 1611 | 0.9725 | 3.8074 | 0.0832 |
| CE5-1 | 16384 | 1132 | 1814 | 1790 | 0.9676 | 4.3173 | 0.0546 |
| CE6-1 | 16896 | 1186 | 1997 | 1872 | 0.9657 | 4.1268 | 0.0666 |
| CE7-1 | 16350 | 1103 | 2501 | 1795 | 0.965 | 3.9040 | 0.0810 |
| CE7-3 | 18122 | 1258 | 1958 | 1999 | 0.9686 | 4.3776 | 0.0531 |
| Non-polluted cloud episodes | | | | | | | |
| CE1-3 | 17662 | 1173 | 1689 | 1687 | 0.9732 | 4.7067 | 0.0327 |
| CE7-2 | 18252 | 1150 | 1732 | 1673 | 0.9729 | 4.3709 | 0.0426 |
| Aerosol (Katra et al., 2014) | 4020 | 1412 | | 2142 | 0.8300 | | |
| Bioaerosol (Madsen et al., 2015) | | | | | | 2.64-3.05 | 0.816-0.922 |
| Rain water in July (Cho & Jang, 2014) | 3055 | 1542 | 13083 | 6387 | | | |
| PM$_{2.5}$ in summer (Franzetti et al., 2011) | | 2222 | | 4036 | | | |
| TSP annual (Bertolini et al., 2013) | 271587 | 765-26,187 | | 107 | | 2.40 | |

The diversity indexes including OTUs, ace, Chao1, coverage, shannon, simpson were defined at 97% sequence similarity.

CE refers to cloud episodes. TSP refers to the total suspended particulate matter.

**Table 3 The identified bacterial species in cloud water samples correlation with the potential ecological function**

| Genus | Identified species | Habitats | Ecological role | Reference |
|---|---|---|---|---|
| *Acinetobacter* [GP] | *A. schindleri* | soil/water | CNN or IN; Opportunistic pathogens | (Mortazavi et al., 2008; Nemec et al., 2001) |
| *Bacillus* [FR] | *B. anthracis* | soil/water/air | CNN or IN; Opportunistic pathogens | (Makino & Cheun, 2003; Mortazavi et al., 2008) |
| *Brevundimonas* [BP] | *B. diminuta* | soil/water | CNN | (Bauer et al., 2003; Han & Andrade, 2005) |
| | *B. vesicularis* | soil/water | Opportunistic pathogens | (Gilad et al., 2009) |
| *Caulobacter* [AP] | *Caulobacter. sp.* | water | Metabolism/Biodegradation | (Nakamura et al., 2007) |
| *Chryseobacterium* [BA] | *C. aquaticum* | soil/water | Protect and promote plants growth | (Gandhi et al., 2009) |
| | *C. jejuense* | soil/water | | (Ben Abdeljalil & Vallance, 2016) |
| *Clostridium* [FR] | *C. tertium* | soil/gut | Opportunistic pathogens | (Miller et al., 2001) |
| *Comamonas* [BP] | *C. testosteroni* | soil/water | Metabolism/Biodegradation | (Goyal & Zylstra, 1996) |
| *Cyanobacterium* [CY] | *Cyanobacterium sp.* | soil/water | Carbon and nitrogen fixing | (Jha et al., 2004) |
| *Deinococcus* [DT] | *D. aquatilis* | soil/water | Extremophiles, radiation-resistant | (Kämpfer et al., 2009) |
| *Delftia* [BP] | *D. tsuruhatensis* | soil/water | Metabolism/Biodegradation | (Geng et al., 2009) |
| *Empedobacter* [BA] | *E. brevis* | soil/water/plant | Opportunistic pathogens | (Bottone et al., 1992) |
| *Methylobacterium* [AP] | *M. aquaticum* | water | Methylotrophic, carbon fixing | (Gallego et al., 2005) |
| | *M. adhaesivum* | soil/water | | (Gallego et al., 2006) |
| *Moraxella* [GP] | *M. osloensis* | soil/animal | Opportunistic pathogens | (Banks et al., 2007) |
| *Novosphingobium* [AP] | *N. aromaticivorans* | soil/water | Metabolism/Biodegradation | (Bell & Wong, 2007) |
| *Staphylococcus* [GP] | *S. equorum* | soil/water/clinic | Opportunistic pathogens | (Nováková et al., 2006) |
| *Phyllobacterium* [AP] | *P. myrsinacearum* | soil/plant | Rhizosphere bacteria, nitrogen fixation | (Gonzalezbashan et al., 2000) |
| *Pseudomonas* [GP] | *P. psychrotolerans* | soil/water | Extremophiles, psychrotolerant | (Hauser et al., 2004) |
| | *P. geniculate* | soil/water/plant | Metabolism/Biodegradation | (Gopalakrishnan et al., 2015; Liu et al., 2014) |
| | *P. putida* | water/soil | Protect and promote plants growth | (Meziane et al., 2005; Reardon et al., 2000) |

| Genus | Identified species | Habitats | Ecological role | Reference |
|---|---|---|---|---|
| | | | CNN or IN | (Amato et al., 2015; Joly et al., 2013) |
| *Rhodococcus* [AC] | *R. ruber* | soil/water | Metabolism/Biodegradation | (Bock et al., 1996) |
| *Sphingomonas* [AP] | *S. faeni* | soil/water | CNN or IN; psychrotolerant | (Ponder et al., 2005) |
| | *S. kaistensis* | soil/water | Metabolism/Biodegradation | (Busse et al., 2003) |
| | *S. leidyi* | soil/water | | (Glaeser & Kämpfer, 2014) |
| *Stenotrophomonas* [GP] | *S. rhizophila* | soil/water/plant | CNN or IN; Rhizosphere bacteria | (Mortazavi et al., 2008; Wolf et al., 2002) |

CNN and IN refers to the bacteria participating in the formation of clouds or rain by acting as cloud condensation nuclei (CNN) and ice nuclei (IN).

Abbreviates are as followed: AP, Alphaproteobacteria; BP, Betaproteobacteria; GP, Gammaproteobacteria; AC, Actinobacteria;

BA, Bacteroidetes; CY, Cyanobacteria; DT, Deinococcus-Thermus; FR, Firmicutes.

5        Biodegradation refers to the bacteria associated with the biodegradation of organic compounds, even toxic pollutants, e.g., aromatic compounds.