# Peer review of "Characteristics of bacterial community in cloud water at Mt. Tai: similarity and disparity under polluted and non-polluted cloud episodes"

_Atmospheric Chemistry and Physics, 2016_

## Referee Comment (RC1) · Anonymous Referee #1 · 21 Nov 2016

The authors have performed taxonomic analysis of the metagenomes from polluted and non-polluted episodes of fog events at Mount Tai. The study has described the key differences in the bacterial composition and associated it with the environmental factors. The results are interesting, however, several key points need to be addresses:

1. The authors performed a 16S analysis of the microbial community which is an excellent and reliable approach to classify the composition. However, the functionality of microbes cannot be assumed on this base. It would be other approaches (e.g. metagenome-assembly based analysis). Therefore, the authors should modify the text accordingly i.e. by not assuming the functional diversity of fog microbes. Please refer to Jiang et. al, Nat Protoc, 2015 for general methodology.

2. Quality of English needs to be revised. Grammatical mistakes and use of wrong terminologies is seen throughout the manuscript. For instance, grammatical mistake in introduction (Line 5), Page 9 line 33 ("... server infections") and Page 10 line 5 ("Previous studies has shown ..."). The author names also have discrepancy in main text and supplement.

3. Few major statements are not supported by references e.g. the authors mentioned that the recruitment of bacteria from various sources to air is harmful to humans upon deposition back to land via fog.

4. Figure captions need to a bit more descriptive/proof read.

---

## Referee Comment (RC2) · Anonymous Referee #2 · 24 Nov 2016

The authors investigated the differences in bacterial community structures from fog wa-
ter droplet samples collected from Mt. Tai in North Plain of China including those clear
and polluted days in July and August of 2014. They performed sequence analysis of
the samples, and also investigated the effects of environmental factors on the bacte-
rial community structure. Overall, it is interesting to study the bacteria in the fog water
samples, especially in higher altitude from a ground. The information developed is use-
ful to understanding the microbial transport and possible roles in atmospheric pollutant
transformation. The authors provided a number of different analyses of their results
and derived some valuable information. Nonetheless, this reviewer does observe the
following drawbacks that need the authors' attention:

[Figure]

1. From their work, it seems they only had one day with higher PM2.5 pollution level, i.e., exceeding 100 ug/mˆ3, and they had more samples from clear days with much lower PM2.5 levels. In their work, they compared them and further derived relevant information. I think the authors have to carefully make their conclusions regarding their limited set of data from a single polluted day. Probably, they can use the 24-hour backward trajectories to discuss more on them.

2. It seems they did not clearly define what level of PM2.5 for which a day can be classified as a polluted day in their method section. Also they should clearly define what those symbols such as "FE" stand for? although I guess it should be "Fog Episode", but they should appear in all figure captions so that readers can easily understand the figures. They should describe that the characteristics of each Fog Episode are shown in relevant Tables in each Figure.

3. be aware that they only performed genus level sequence and they cannot derive any particular bacterial species, especially when they discuss about pathogens. For certain genera, not all of their species are pathogens or opportunistic pathogens.

4. I did not see any concentration levels for the total bacteria in their fog water droplet samples? Did they perform qPCR for total bacteria for their samples?

5. It would be great if they can provide data for fungal spores. I guess there will be some fungal spores in the fog water droplets.

6. For certain bacteria, when they are stored at 4 degree C, they can still grow. How long did it elapse between the collection and their actual analysis?

---

## Author Comment (AC1) · 21 Dec 2016

**Characteristics of bacterial community in fog water at Mt. Tai: similarity and disparity under polluted and non-polluted fog episodes**

Min Wei [1], Caihong Xu [1], Jianmin Chen [1,2,*], Chao Zhu [1], Jiarong Li [1], Ganglin Lv [1]

[1] Environment Research Institute, School of 5 Environmental Science and Engineering, Shandong University, Ji'nan 250100, China

[2] Shanghai Key Laboratory of Atmospheric Particle Pollution and Prevention (LAP), Fudan Tyndall Centre, Department of Environmental Science & Engineering, Fudan University, Shanghai 200433, China

*Correspondence to*: JM.Chen (jmchen@sdu.edu.cn)

**Response to reviewer 2**

The authors investigated the differences in bacterial community structures from fog water droplet samples collected from Mt. Tai in North Plain of China including those clear and polluted days in July and August of 2014. They performed sequence analysis of the samples, and also investigated the effects of environmental factors on the bacterial community structure. Overall, it is interesting to study the bacteria in the fog water samples, especially in higher altitude from a ground. The information developed is useful to understanding the microbial transport and possible roles in atmospheric pollutant transformation. The authors provided a number of different analyses of their results and derived some valuable information. Nonetheless, this reviewer does observe the following drawbacks that need the authors' attention:

We thank the reviewer for the beneficial comments on our manuscript. We respond to the reviewer comments in detail below. The responses to reviewer are in red.

**1.** From their work, it seems they only had one day with higher $PM_{2.5}$ pollution level, i.e., exceeding 100 ug/m$^3$, and they had more samples from clear days with much lower $PM_{2.5}$ levels. In their work, they compared them and further derived relevant information. I think the authors have to carefully make their conclusions regarding their limited set of data from a single polluted day. Probably, they can use the 24-hour backward trajectories to discuss more on them.

**Response of the authors:** According to our field observation data on the summit of Mt.Tai in the year of 2014 and 2015(unpublished), the 24 h average $PM_{2.5}$ mass concentration was basically less than 100 ug/m$^3$, sometimes with relative lower concentration less than 10 ug/m$^3$. The possible reason was that the Mt. Tai was the

highest mountain in North China Plain (1534 m a.s.l), which was typically as the background of atmospheric quality. The $PM_{2.5}$ was relatively low than other regions in the North China Plain. Similar results were obtained by other studies and suggest the relation of $PM_{2.5}$ and attitude. Gehrig and Buchmann studied the seasonal variations and spatial distribution of ambient $PM_{10}$ and $PM_{2.5}$ concentrations. In comparison to other study area (different attitude), the lowest $PM_{2.5}$ concentrations were observed at the elevated site Chaumont (1140 m a.s.l.) (Gehrig & Buchmann, 2003). Similarly, Fan et al studied the vertical distribution of $PM_{2.5}$ concentration in fog and haze days in Beijing and suggest that $PM_{2.5}$ concentrations decreased with the increase of altitude (Fan et al., 2009).

**The 24 h average $PM_{2.5}$ mass concentration according to our field observation data**

| 2014 | PM2.5 ($\mu g/m^3$) | 2014 | PM2.5 ($\mu g/m^3$) | 2015 | PM2.5 ($\mu g/m^3$) | 2015 | PM2.5 ($\mu g/m^3$) |
|---|---|---|---|---|---|---|---|
| 2014/7/23 | 53.9 | 2014/8/9 | 30.1 | 2015/7/6 | 62.3 | 2015/7/23 | 34.8 |
| 2014/7/24 | 49.9 | 2014/8/10 | 67.5 | 2015/7/7 | 64.8 | 2015/7/24 | 34.2 |
| 2014/7/25 | 12.2 | 2014/8/11 | 51.1 | 2015/7/8 | 112 | 2015/7/25 | 36 |
| 2014/7/26 | 44.5 | 2014/8/12 | 45.5 | 2015/7/9 | 71.6 | 2015/7/26 | 33.7 |
| 2014/7/27 | 66.9 | 2014/8/13 | 47.4 | 2015/7/10 | 61.2 | 2015/7/27 | 65.6 |
| 2014/7/28 | 97.5 | 2014/8/14 | 49.3 | 2015/7/11 | 80.6 | 2015/7/28 | 49 |
| 2014/7/29 | 73.4 | 2014/8/15 | 47.6 | 2015/7/12 | 31.4 | 2015/7/29 | 7.9 |
| 2014/7/30 | 23.4 | 2014/8/16 | 66.2 | 2015/7/13 | 57 | 2015/7/30 | 12.6 |
| 2014/7/31 | 56.2 | 2014/8/17 | 69.5 | 2015/7/14 | 53.5 | 2015/7/31 | 14.9 |
| 2014/8/1 | 17.7 | 2014/8/18 | 53.7 | 2015/7/15 | 52.1 | 2015/8/1 | 17.2 |
| 2014/8/2 | 42.5 | 2014/8/19 | 64.9 | 2015/7/16 | 54.5 | 2015/8/2 | 35 |
| 2014/8/3 | 45.5 | 2014/8/20 | 62.1 | 2015/7/17 | 62.9 | 2015/8/3 | 6.1 |
| 2014/8/4 | 86.5 | 2014/8/21 | 71.3 | 2015/7/18 | 51.9 | 2015/8/4 | 5.9 |
| 2014/8/5 | 54.8 | 2014/8/22 | 54.2 | 2015/7/19 | 44.4 | 2015/8/5 | 9.9 |
| 2014/8/6 | 18.2 | 2014/8/23 | 48.8 | 2015/7/20 | 21.4 | 2015/8/6 | 19.3 |
| 2014/8/7 | 44.1 | 2014/8/23 | 42.8 | 2015/7/21 | 43.1 | 2015/8/7 | 7.8 |
| 2014/8/8 | 40.3 | | | 2015/7/22 | 55.8 | 2015/8/8 | 26.2 |

In addition, the listed $PM_{2.5}$ concentration in Table 1 was the average value during a fog process, not the 24 h average concentration. The 24 h $PM_{2.5}$ concentration in fog days was lower than non-fog days which possible due to the wet deposition. During fog episodes, $PM_{2.5}$ concentration varied with fog process. The mass concentration was high in the initiation of fog episode, with the development and dissipation of fog, the concentration steadily reduced due to the reduced input (nighttime) and wet deposition.

In the present study, the polluted fog episodes were defined according to the 24 h concentration of WHO air quality guideline (25 $ug/m^3$) and the standard was applied

by Australia, New Zealand and European Union.

In the section of 3.4, we have discussed the influence of air mass and meteorological conditions on PM2.5. The Sampling site was 1534 m a.s.l, air pollution was typically effected by air mass over long term transport than local emissions.We use the 24-hour backward trajectories to track the air mass and combined the wind direction and wind speed to deeply discuss the possible driven factors. The main points obtained was that air mass from the contaminated area through long term transport with lower wind speeds, largely reduce the diffusion rate of pollutants and thus lead to the sustained high $PM_{2.5}$ during polluted fog episodes.

**2.** It seems they did not clearly define what level of PM2.5 for which a day can be classified as a polluted day in their method section. Also they should clearly define what those symbols such as "FE" stand for? although I guess it should be "Fog Episode", but they should appear in all figure captions so that readers can easily understand the figures. They should describe that the characteristics of each Fog Episode are shown in relevant Tables in each Figure.

**Response of the authors:** thank you to your suggestion, we have clearly define the polluted fog episodes and indicated the abbreviation in the table and figures. The polluted fog episodes were defined according to the 24 h concentration of WHO air quality guideline (25 ug/m$^3$) and the standard was applied by Australia, New Zealand and European Union. During a fog episode, the average $PM_{2.5}$ concentration higher than 25 ug/m$^3$ was classified as polluted. WHO proposes $PM_{2.5}$ less than 10 ug/m$^3$ is safe. Elevated $PM_{2.5}$ concentration will highly increase health risks. The high pollutant and pathogens are detrimental to individuals (Fang et al., 2013). $PM_{2.5}$ concentrations were compared to the 24 h World Health Organization limit of 25 ug/m$^3$.

**3.** be aware that they only performed genus level sequence and they cannot derive any particular bacterial species, especially when they discuss about pathogens. For certain genera, not all of their species are pathogens or opportunistic pathogens.

**Response of the authors:** we have revised the discussion about potential pathogens. Yes, the Miseq sequencing identify bacterial taxa mostly at the genus level. In the present study, the V3-V4 hypervariable region of the bacterial 16S rRNA gene was amplified. Single OTUs were removed and taxonomy was assigned to each OTU using the Ribosomal Database Project (RDP) classifier in QIIME, with a minimum confidence cutoff of 80% against the Silva reference database (silva 119,http://www.arb-silva.de/) to the genus level. Finally, we focused on those

bacterial genera that included species known or suspected to be opportunistic pathogens. To this aim, we performed a systematic literature review to identify potential pathogenic bacteria in water habitats (Bibby et al., 2010; Guo & Zhang, 2012; Luo & Angelidaki, 2014).

Previous studies have discussed the bacterial pathogens based on NGS sequencing (454 pyrosequencing, Miseq, Ion Torrent PGM). Razzauti et al conducted a comparison between transcriptome sequencing and 16S metagenomics for detection of bacterial pathogens in Wildlife (Razzauti et al., 2015) and suggest that 16S approach was able to determine bacterial diversity in each individual. They also indicated that NGS techniques (454-pyrosequencing and MiSeq) are very affordable candidates and could become routine approaches in future large-scale epidemiological studies. Luo and Angelidaki studied the bacterial communities and bacterial pathogens with the high sequencing depth by Ion Torrent PGM (16S rRNA gene sequencing), they suggest the Ion Torrent PGM is also possible to detect the potential bacterial pathogens in biogas reactors. To identify potential pathogens, they use the reference bacterial pathogen database and identified the potential bacterial pathogens at the species level (Luo & Angelidaki, 2014).

**4.** I did not see any concentration levels for the total bacteria in their fog water droplet samples? Did they perform qPCR for total bacteria for their samples?

**Response of the authors:** Due to the complexity of fog water collection, the amount for each fog episode ranged from 40 to 200 mL based on the fog duration, mist or dense fog. For the majority fog episodes, e.g. FE1-3, FE2-1, FE4-1, FE4-2, the remained volume was inadequate for other analysis after Miseq sequencing.

The collected fog water samples were processed by genomic DNA extracting, PCR amplification, Miseq sequencing and qPCR. In DNA extraction, some samples DNA cannot be successfully extracted and require repeated extraction, thus consume more sample volume. We have performed qPCR for total bacteria after Miseq. However, after miseq, no remaining sample DNA for the further analysis for certain samples. QPCR was just performed for the samples with sufficient DNA and bacterial concentration are listed in the following table. Therefore, we did not discuss the total bacterial concentration in the manuscript.

Bacterial concentration for different fog episodes

| Sample | Collected volume (mL) | Bacterial concentration (cells/mL) |
|---|---|---|
| FE1-1 | 90 | $8.9 \times 10^4$ |
| FE1-2 | 80 | $1.3 \times 10^5$ |

| | | |
|---|---|---|
| FE1-3 | 55 | Not detected |
| FE2-1 | 75 | Not detected |
| FE3-1 | 100 | Not detected |
| FE4-1 | 65 | Not detected |
| FE4-2 | 40 | Not detected |
| FE4-3 | 40 | Not detected |
| FE5-1 | 50 | Not detected |
| FE6-1 | 60 | Not detected |
| FE7-1 | 210 | $1.5 \times 10^5$ |
| FE7-2 | 200 | $5.8 \times 10^4$ |
| FE7-3 | 120 | $1.6 \times 10^5$ |

**5.** It would be great if they can provide data for fungal spores. I guess there will be some fungal spores in the fog water droplets.

**Response of the authors:** We agree that the investigations of fungal diversity in fog water are very important areas for future work. Your suggestions are very helpful for our further study. However, the analysis of fungal spores requires substantial amounts of additional work, including the resequencing and culture-dependent experiments. The remaining parts of the samples are unable to support the above experiments. We therefore decide not to include these in our current manuscript and leave for future work. The next studies on microbial community will consider the fungal diversity in fog water and other aerosol samples.

**6.** For certain bacteria, when they are stored at 4 degree C, they can still grow. How long did it elapse between the collection and their actual analysis?

**Response of the authors:** thank you for your comments. We have modified the description and clearly described the storage conditions of the sample for different measurements. Basal analysis of water typically included chemical and biological two parts. For chemical analysis, part of samples were stored in pre-baked glass bottles, immediately preserved with hydrochloric acid (HCl, pH <2.0), stored at 4 ℃ in ice box during transit, and analyzed upon arrival at the Laboratory. Samples for microbial diversity analysis were not preserved with hydrochloric acid and stored with dry ice in transit, and frozen at -80 ℃ until further analysis.

**Reference**

Bibby, K., Viau, E., Peccia, J.: Bibby K, Viau E, Peccia J.. Pyrosequencing of the 16S rRNA gene

to reveal bacterial pathogen diversity in biosolids. Water Res 44: 4252-4260, Water. Res., 44, 4252-4260, 2010.

Fan, W., Hu, B., Wang, Y., Wang, S., Sun, Y.: Measurements on the Vertical Distribution of $PM_{2.5}$ Concentration in Fog and Haze Days in Beijing City, Climatic & Environmental Research (Chinese) 14, 631-638, 2009.

Fang, W., Yang, Y., Xu, Z.: $PM_{10}$ and $PM_{2.5}$ and health risk assessment for heavy metals in a typical factory for cathode ray tube television recycling, Environ. Sci. Technol., 47, 12469-12476, 2013.

Gehrig, R., Buchmann, B.: Characterising seasonal variations and spatial distribution of ambient $PM_{10}$ and $PM_{2.5}$ concentrations based on long-term Swiss monitoring data, Atmos. Environ., 37, 2571-2580, 2003.

Guo, F., Zhang, T.: Profiling bulking and foaming bacteria in activated sludge by high throughput sequencing, Water. Res., 46, 2772-2782, 2012.

Luo, G., Angelidaki, I.: Analysis of bacterial communities and bacterial pathogens in a biogas plant by the combination of ethidium monoazide, PCR and Ion Torrent sequencing, Water. Res., 60, 156-163, 10.1016/j.watres.2014.04.047, 2014.

Razzauti, M., Galan, M., Bernard, M., Maman, S., Klopp, C., Charbonnel, N., et al.: A Comparison between Transcriptome Sequencing and 16S Metagenomics for Detection of Bacterial Pathogens in Wildlife, Plos. Neglect. Trop. D., 9, 876-887, 2015.

---

## Author Comment (AC2) · 21 Dec 2016

**Characteristics of bacterial community in fog water at Mt. Tai: similarity and disparity under polluted and non-polluted fog episodes**

Min Wei [1], Caihong Xu [1], Jianmin Chen [1,2,*], Chao Zhu [1], Jiarong Li [1], Ganglin Lv [1]

[1] Environment Research Institute, School of 5 Environmental Science and Engineering, Shandong University, Ji'nan 250100, China

[2] Shanghai Key Laboratory of Atmospheric Particle Pollution and Prevention (LAP), Fudan Tyndall Centre, Department of Environmental Science & Engineering, Fudan University, Shanghai 200433, China

*Correspondence to*: JM.Chen (jmchen@sdu.edu.cn)

**Response to reviewer 1**

The authors have performed taxonomic analysis of the metagenomes from polluted and non-polluted episodes of fog events at Mount Tai. The study has described the key differences in the bacterial composition and associated it with the environmental factors. The results are interesting, however, several key points need to be addresses:

We thank the reviewer for the beneficial comments on our manuscript. We respond to the reviewer comments in detail below. The responses to reviewer are in red.

**1.** The authors performed a 16S analysis of the microbial community which is an excellent and reliable approach to classify the composition. However, the functionality of microbes cannot be assumed on this base. It would be other approaches (e.g. metagenome-assembly based analysis). Therefore, the authors should modify the text accordingly i.e. by not assuming the functional diversity of fog microbes. Please refer to Jiang et. al, Nat Protoc, 2015 for general methodology.

**Response of the authors:** thank you for your comments.
First, yes, the Miseq 16S rRNA gene sequencing was a powerful tool in microbial diversity investigation, which provides a comprehensive understanding of community composition. The metagenome-assembly based analysis have been widely used in microbial community functional analysis, e.g., Cao et al. (2014) described the microbial communities in $PM_{2.5}$ and $PM_{10}$ using metagenomics during a serious smog event (Cao et al., 2014), Be et al examined aerosolized microorganisms in urban airborne microbes and revealed the metagenomic complexity of urban aerosols and the potential of genomic analytical techniques for biosurveillance and monitoring of threats to public health (Be et al., 2015). We also studied the suggested reference

Jiang et. al for general methodology (Jiang et al., 2015). However, due to the complexity of fog water collection, the amount for each fog episode ranged from 40 to 200 mL based on the duration, mist or dense fog. The sampled volume was inadequate for metagenomic analysis.

Second, community functions are based on community composition, bacterial taxa. For specific functional bacteria, e.g. Rhizobia (*Phyllobacterium myrsinacearum*) are involved in Biological nitrogen fixation, and favorable for plant growth, Methanotrophic Bacteria (*Methylobacterium aquaticum*, *Methylobacterium adhaesivum*) are related to methane oxidation and carbon recycling. We have added the references attached to the bacterial species in Table 3.

Third, to acquire more accurate community function, we performed PICRUST function prediction in the revised manuscript. Phylogenetic Investigation of Communities by Reconstruction of Unobserved States (PICRUST) can be used to predict the metabolic function spectrum of corresponding bacteria and archaea based on the 16S rRNA gene sequence (Langille et al., 2013). PICRUST has been used in bacterial diversity and function analysis (Corrigan et al., 2015; Wu et al., 2016) and it was applied in community function prediction in the present study. The predicted function including metabolism and human disease are essential part of atmospheric microbial community function, which are likely attributed to the bacterial gene content from the identified species in Table 3.

[Figure]

**Predictive functional profiling of microbial communities, using 16s rRNA gene sequences**

**2.** Quality of English needs to be revised. Grammatical mistakes and use of wrong terminologies is seen throughout the manuscript. For instance, grammatical mistake in introduction (Line 5), Page 9 line 33 (": : : server infections") and Page 10 line 5 ("Previous studies has shown : : :"). The author names also have discrepancy in main text and supplement.

**Response of the authors:** We have polished the manuscript with a professional assistance in writing. The mistakes have been corrected.

**3.** Few major statements are not supported by references e.g. the authors mentioned that the recruitment of bacteria from various sources to air is harmful to humans upon deposition back to land via fog.

**Response of the authors:** thank you for your comments.

Numerous studies are focused on the potential pathogens identified in atmospheric particulate matter ($PM_{2.5}$ and $PM_{10}$) (Cao et al., 2014; Creamean et al., 2013), rain water (Kaushik & Balasubramanian, 2012; Simmons et al., 2001), and indicated that health risk-related bacteria in atmospheric samples should be concerned. For fog water, studies of health risks to individuals are typically focused on the chemical characteristic, e.g. the low pH (acid fog) (Hackney et al., 1989), PAH (Ehrenhauser et al., 2012), etc. Limited literatures discussed the microorganism in fog/cloud water suggested the potential pathogens in fog/cloud water(Vaïlingom et al., 2012), they find potential plant pathogens such as *Pseudomonas syringae* and *Xanthomonas campestris.* In their study, they suggest that the wet deposition play a major role in the dispersion of microorganisms, which appear as an extension of the phyllosphere and carry living species of plant pathogens that could then infect new hosts through precipitation.

In the present study, potential human pathogens were identified in the fog water samples. However, the detailed health risks should be prudently assessed. Further study depending on the culture-dependent method and biochemical experiments will perform to check the pathogenicity. We have revised the relevant discussion.

**4.** Figure captions need to a bit more descriptive/proof read.

**Response of the authors:** More detailed descriptions have been added to the figure captions.

**Figure 1** Bacterial community variation for the fog episodes at the phylum (A) and class (B) level. FE refers to the fog episodes. Bar graphs for each sample represent the percentage of taxa assigned to each phylum with 80% bootstrap confidence.

Taxonomic summary of the most abundant taxa (more than 1%) across all fog samples are indicated in the bar graphs.

**Figure 2** Bacterial taxa are related to KEGG functional pathways. Bacterial gene functions were predicted from 16S rRNA gene-based microbial compositions using the PICRUSt algorithm to make inferences from KEGG annotated databases. Spearman's correlation coefficients were estimated for each pairwise comparison of genus counts and KEGG pathway counts. Selected KEGG pathways relating to metabolism and disease infection and predominant genera are included in the heatmap. Red color refers to the positive correlation, and green indicates a negative correlation. Correlation is significant at *P < 0.05, **P < 0.01.

**Figure 3** Schematic representation of bioaerosols life cycle and potential influence on atmosphere, ecosystem and human health, modified from Poeschl (Poeschl, 2006). The predominant identified bacterial species with potential ecological functions are indicated in the figure. Bioaerosols emissions and resuspension from various terrestrial environments, e.g., soil, water, plants, animals or human beings, may include pathogenic or functional species. These bacteria can be attached to particles or incorporated into water droplets of clouds/fog. Certain species can serve as biogenic nuclei for Cloud Condensation Nuclei (CCN) and Ice Nuclei (IN), which induce rain formation, precipitation, and wet deposition of gases and particles. For the potential pathogens and functional bacteria, during fog process, they can be deposited back to land via deposition and possibly induce infections to human health and impose effect on the diversity and function of aquatic and terrestrial ecosystems.

**Figure 4** Bacterial taxa significantly differentiated between the polluted and non-polluted fog episodes identified by linear discriminant analysis coupled with effect size (LEfSe). The LDA effect sizes (left) were calculated using the default parameters. The taxonomic cladogram (right) with LDA values higher than 3.5 comparing all bacterial taxa and significantly discriminant taxon nodes are colored and branch areas are shaded according to the highest-ranked variety for that taxon. Taxa with significant difference in polluted and non-polluted fog episodes are indicated in red and green circles, respectively. Bacterial taxa with nonsignificant differences are represented as yellow circles and the diameter of the circles are proportional to relative abundance.

**Figure 5** Biplot of the environmental variables and genus-level community structure using a redundancy analysis (RDA) model, describing the variation in bacterial community explained by environmental variable. Species data are listed in Table S2. The selected environmental variables are significant (P < 0.05) using Monte Carlo permutation testing. Species are labeled with triangle, the closer of which indicates existence in similar environment. Environmental variables are showed by arrows; the relative length is positive correlation with the importance in influencing bacterial

community structure. The angle between the arrow and the ordination axis suggest the variable response respect to the RDA gradient. The axes explain 73.3% of the variability in the data for bacteria and $PM_{2.5}$ seems to be the most important environmental variable shaping the bacterial community.

**Figure 6** Air mass transport pathways for each fog episodes using the Hybrid Single Particle Lagrangian Integrated Trajectory (HYSPLIT) model. 24-hour backward trajectories were calculated for air parcels arriving at the summit of Mt. Tai ($36^{o}18'$ N, $117^{o}13'$ E, and 1534 m a.s.l). The polluted fog episodes are indicated in red lines, and green lines are non-polluted fog episodes. Wind Rose Diagram to quantitative analysis of wind speed and wind direction during sampling time. The frequency of winds is plotted by wind direction, the color bands show wind speed ranges. The direction of the longest spoke shows the wind direction with the greatest frequency.

**Reference**

Be, N.A., Thissen, J.B., Fofanov, V.Y., Allen, J.E., Rojas, M., Golovko, G., et al.: Metagenomic analysis of the airborne environment in urban spaces, Microb. Ecol., 69, 346-355, 10.1007/s00248-014-0517-z, 2015.

Cao, C., Jiang, W., Wang, B., Fang, J., Lang, J., Tian, G., et al.: Inhalable microorganisms in Beijing's $PM_{2.5}$ and $PM_{10}$ pollutants during a severe smog event, Environ. Sci. Technol., 48, 1499-1507, 2014.

Corrigan, A., de Leeuw, M., Penaud-Frezet, S., Dimova, D., Murphy, R.A.: Phylogenetic and functional alterations in bacterial community compositions in broiler ceca as a result of mannan oligosaccharide supplementation, Appl. Environ. Microbiol., 81, 3460-3470, 10.1128/AEM.04194-14, 2015.

Creamean, J.M., Suski, K.J., Rosenfeld, D., Cazorla, A., Demott, P.J., Sullivan, R.C., et al.: Dust and biological aerosols from the Sahara and Asia influence precipitation in the western U.S, Science, 339, 1572-8, 2013.

Ehrenhauser, F.S., Khadapkar, K., Wang, Y., Hutchings, J.W., Delhomme, O., Kommalapati, R.R., et al.: Processing of atmospheric polycyclic aromatic hydrocarbons by fog in an urban environment, J. Environ. Monitor., 14, 2566-2579, 2012.

Hackney, J.D., Linn, W.S., Avol, E.L.: Acid fog: effects on respiratory function and symptoms in healthy and asthmatic volunteers, Environ. Health Perspect., 79, 159-162, 1989.

Jiang, W., Liang, P., Wang, B., Fang, J., Lang, J., Tian, G., et al.: Optimized DNA extraction and metagenomic sequencing of airborne microbial communities, Nat. Protoc., 10, 768-779, 10.1038/nprot.2015.046, 2015.

Kaushik, R., Balasubramanian, R.: Assessment of bacterial pathogens in fresh rainwater and airborne particulate matter using Real-Time PCR, Atmos. Environ., 46, 131-139, 10.1016/j.atmosenv.2011.10.013, 2012.

Langille, M.G., Zaneveld, J., Caporaso, J.G., McDonald, D., Knights, D., Reyes, J.A., et al.: Predictive functional profiling of microbial communities using 16S rRNA marker gene sequences, Nat. Biotechnol., 31, 814-821, 10.1038/nbt.2676, 2013.

Simmons, G., Hope, V., Lewis, G., Whitmore, J., Gao, W.: Contamination of potable

roof-collected rainwater in Auckland, New Zealand, Water. Res., 35, 1518-1524, 2001.

Vaïtilingom, M., Attard, E., Gaiani, N., Sancelme, M., Deguillaume, L., Flossmann, A.I., et al.: Long-term features of cloud microbiology at the puy de Dôme (France), Atmos. Environ., 56, 88-100, 2012.

Wu, J., Peters, B.A., Dominianni, C., Zhang, Y., Pei, Z., Yang, L., et al.: Cigarette smoking and the oral microbiome in a large study of American adults, ISME. J., 10, 2435-2446, 10.1038/ismej.2016.37, 2016.

---

## Author Comment (AC3) · 21 Dec 2016

**The revised manuscript**

[revised manuscript text omitted]
 metagenome predicted from 16S rRNA gene-based microbial compositions using the PICRUSt algorithm, and functional inferences were made against with the Kyoto Encyclopedia of Gene and Genomes (KEGG) annotated databases. Spearman's correlation coefficients were estimated for each pairwise comparison of genus and KEGG pathway counts. Selected KEGG pathways relating to metabolism and disease infection and predominant genera are included in the heatmap. Correlation is significant at P-value of less than 0.05 and 0.01.

Alpha diversity was assessed by examining the rarefaction curves, shannon-wiener curves and rank-abundance curves calculated with Mothur (v.1.34.0; http://www.mothur.org) (Schloss et al., 2009) and visualized in R project (v.3.1.3; https://www.r-project.org/). Community richness estimators including the observed OTUs, nonparametric Chao1, ACE, and community diversity estimators including Shannon and Simpson indexes were also calculated with Mothur. Moreover, the Good's coverage was used to evaluate the sequencing depth.

Principal component analysis (PCA) was carried out to visualize the changes in bacterial community between polluted and non-polluted fog samples. The PCA plots were constructed based on Bray-Curtis similarity index calculated with the abundance of OTUs using the BIODIVERSITYR package (Kindt & Coe, 2005) in R. The difference in OTU composition for samples collected in polluted and non-polluted fog episodes was tested by the analysis of similarity (ANOSIM) (Clarke, 1993). ANOSIM was performed with the VEGAN package in R. Linear discriminant analysis effect size (LEfSe, http://www.huttenhower.sph.harvard.edu/galaxy/) was applied to identify differentially abundant bacterial taxa associated with the polluted and non-polluted

fog episodes at genus or higher taxonomy levels (Segata et al., 2011). All statistical tests were two-sided, and a P-value of less than 0.05 was considered statistically significant.

[revised manuscript text omitted]
 diversity variability of bacterial community structure and strongly correlated with represent bacterial genera. Bacterial community composition was highly variable under $PM_{2.5}$ mass concentration in this study, which was consistent with previous studies that $PM_{2.5}$ was important environmental factor shaping the variation of community composition (Cao et al., 2014). Moreover, statistical analysis, e.g., correlation or multiple linear regression, indicated that $PM_{2.5}$ exhibited a negative correlation with airborne bacteria in haze days (Gandolfi et al., 2015; Gao et al., 2015), whereas in another study, spearman correlation analysis showed $PM_{2.5}$ exhibited a significant positive correlations with the airborne microbe concentration (Dong et al., 2016). Previous study has suggested that nutrients are related to the distribution of bacteria in water habitats (Newton et al., 2011). Possibly, the inorganic and organic compounds in particulate matter ($PM_{2.5}$) can be available nutrients for microbial growth. However, aggregation of harmful substances such as heavy metals, polycyclic aromatic hydrocarbons would be toxic for bacteria under high $PM_{2.5}$ mass concentration. Since the $PM_{2.5}$'s two-sided influences on bacterial growth, bacterial community from polluted and non-polluted samples was significantly correlated with $PM_{2.5}$ mass concentration.

[revised manuscript text omitted]
 from 16S rRNA gene-based microbial compositions using the PICRUSt algorithm to make inferences from KEGG annotated databases. Spearman's correlation coefficients were estimated for each pairwise comparison of genus and KEGG pathway counts. Selected KEGG pathways relating to metabolism and disease infection and predominant genera are included in the heatmap. Red color refers to the positive correlation, and green indicates a negative correlation. Correlation is significant at *P < 0.05, **P < 0.01.

[Figure]

**Figure 3** Schematic representation of bioaerosols life cycle and potential influence on atmosphere, ecosystem and human health, modified from Poeschl (Poeschl, 2006). The predominant bacteria species with potential functions are indicated in the figure.

5 Bioaerosols emissions and resuspension from various terrestrial environments, e.g., soil, water, plants, animals or human beings, may include pathogenic or functional species. These bacteria can be attached to particles or incorporated into water droplets of clouds/fog. Certain species can serve as biogenic nuclei for Cloud Condensation Nuclei (CCN) and Ice Nuclei (IN), which induce rain formation, precipitation, and

10 wet deposition of gases and particles. For the potential pathogens and functional bacteria, during fog process, they can be deposited back to land via deposition and possibly induce infections to human health and impose effect on the diversity and function of aquatic and terrestrial ecosystems.

[Figure]

**Figure 4** Bacterial taxa significantly differentiated between the polluted and non-polluted fog episodes identified by linear discriminant analysis coupled with effect size (LEfSe). The LDA effect sizes (left) were calculated using the default parameters. The taxonomic cladogram (right) with LDA values higher than 3.5 comparing all bacterial taxa and significantly discriminant taxon nodes are colored and branch areas are shaded according to the highest-ranked variety for that taxon. Taxa with significant difference in polluted and non-polluted fog episodes are indicated in red and green circles, respectively. The bacterial taxa with nonsignificant differences are represented as yellow circles and the diameter of the circles are proportional to relative abundance.

[Figure]

**Figure 5** Biplot of the environmental variables and genus-level community structure using a redundancy analysis (RDA) model, describing the variation in bacterial community explained by environmental variable. FE refers to fog episodes. Polluted fog episodes are indicated in red circle, and non-polluted fog episodes are green squares. Species data are listed in Table S2. The selected environmental variables are significant ($P < 0.05$) using Monte Carlo permutation testing. Species are labeled with triangle, the closer of which indicates existence in similar environment. Environmental variables are showed by arrows; the relative length is positive correlation with the importance in influencing bacterial community structure. The angle between the arrow and the ordination axis suggest the variable response respect to the RDA gradient. The two axes explain 73.3% of the variability. For bacteria, $PM_{2.5}$ seems to be the most important environmental variable shaping the community structure.

[Figure]

**Figure 6** Air mass transport pathways for the fog episodes using the Hybrid Single Particle Lagrangian Integrated Trajectory (HYSPLIT) model. 24-hour backward trajectories were calculated for air parcels arriving at the summit of Mt. Tai (36°18′ N, 117°13′ E, and 1534 m a.s.l). FE refer to fog episodes. The polluted fog episodes are indicated in red lines, and green lines are non-polluted fog episodes. Wind Rose Diagram to quantitative analysis of wind speed and wind direction during sampling time. The frequency of winds is plotted by wind direction, the color bands show wind speed range. The direction of the longest spoke shows the wind direction with the greatest frequency.

**Table 1 Description fog episodes at Mt.Tai, China**

| Fog episodes | Data | Samples | Start time (BJT) | Stop time (BJT) | Duration (h) | $PM_{2.5}$ [a] ($\mu g \cdot m^{-3}$) | $PM_{2.5}$ [b] ($\mu g \cdot m^{-3}$) | pH | EC ($\mu S \cdot cm^{-1}$) | Pollution |
|---|---|---|---|---|---|---|---|---|---|---|
| FE1 | 24 Jul 2014 | FE1-1 | 8:50 | 15:30 | 6:40 | 105.07 | 49.89 | 4.03 | 583 | A |
| | | FE1-2 | 15:30 | 17:30 | 2:00 | 22.35 | | 4.32 | 219.2 | B |
| | | FE1-3 | 17:30 | 22:51 | 5:21 | 14.66 | | 5.74 | 104.4 | B |
| FE2 | 5 Aug 2014 | FE2-1 | 6:45 | 9:17 | 2:32 | 30.36 | 54.84 | 5.80 | 275.7 | A |
| FE3 | 5 Aug - 6 Aug 2014 | FE3-1 | 19:05 | 4:01 | 8:56 | 42.25 | 18.19 | 5.10 | 501 | A |
| FE4 | 14 Aug - 15 Aug 2014 | FE4-1 | 22:41 | 0:44 | 2:03 | 42.69 | 48.47 | 6.36 | 170.4 | A |
| | | FE4-2 | 0:44 | 5:06 | 4:22 | 47.98 | | 5.34 | 86.34 | A |
| | | FE4-3 | 5:06 | 6:03 | 0:57 | 36.88 | | 4.89 | 64.95 | A |
| FE5 | 17 Aug 2014 | FE5-1 | 10:10 | 11:18 | 1:08 | 63.18 | 69.54 | 5.20 | 120.5 | A |
| FE6 | 17 Aug - 18 Aug 2014 | FE6-1 | 22:18 | 1:25 | 3:07 | 54.33 | 53.70 | 3.80 | 321.8 | A |
| FE7 | 23 Aug 2014 | FE7-1 | 2:30 | 4:38 | 2:08 | 30.45 | 48.83 | 4.38 | 356.2 | A |
| | | FE7-2 | 4:38 | 6:21 | 1:43 | 23.39 | | 5.01 | 207.5 | B |
| | | FE7-3 | 6:21 | 9:20 | 2:59 | 41.60 | | 5.74 | 187.6 | A |

FE refers to fog episode. BJT refers to Beijing Time, which equals UTC + 8. EC refers to the electric conductivity.

The A, B refers to the the polluted and non-polluted samples based on the WHO 24-hr average standard $PM_{2.5}$ mass concentration

($PM_{2.5} = 25$ $\mu g \cdot m^{-3}$), respectively. The $PM_{2.5}$ [a] refers to the average value during a fog process.

Daily average concentration is indicated with $PM_{2.5}$ [b].

[revised manuscript text omitted]

Abbreviates are as followed: AP, Alphaproteobacteria; BP, Betaproteobacteria; GP, Gammaproteobacteria; AC, Actinobacteria; BA, Bacteroidetes; CY, Cyanobacteria; DT, Deinococcus-Thermus; FR, Firmicutes. Biodegradation refers to the bacteria associated with the biodegradation of organic compounds, even toxic pollutants, e.g., aromatic compounds.

---

## Referee Comment (RC3) · Anonymous Referee #3 · 23 Dec 2016

Min et al examine bacteria present in cloud water samples collected at Mt. Tai, China. They use a variety of techniques to examine the community composition of bacteria in the samples and attempt to assess differences as a function of a variety of environmental parameters, especially fine particle concentration levels. While the dataset is interesting and the work novel, I have numerous concerns about the work and its presentation.

Major comments:

1. The authors never make it very clear why they are examining bacteria in clouds (they are looking at clouds, not fog – see below). They talk about the importance of interaction with fog, but don't clarify why such interactions are important. They speak

about deposition in clouds, but why is this really important if such bacteria would be deposited anyway by wet or dry processes? Bacteria in cloud drops get there through scavenging of aerosol particles that are either themselves bacteria or have bacteria attached. Why, then, is it important to look at bacteria in cloud water? Why not look at them directly in PM2.5 samples? This would allow a much larger dataset to be examined, which would greatly help statistical analyses of relationships with environmental variables. For example, if one is interested in examining changes in bacterial populations with PM2.5 levels, it would be much more straightforward to look at bacteria directly in PM2.5.

2. One might be interested in examining how cloud processing affects bacteria. For example, do they differentially scavenge and deposit bacteria from a certain subset of aerosol particles? Do the bacteria reproduce in clouds as suggested by Fuzzi? Does interaction with fogs alter the viability of bacteria in some way. The authors do not examine any such questions that would be very relevant to bacteria in fog.

3. I have many concerns about the way in which the authors assess differences in bacteria in fog between polluted and nonpolluted conditions. Chief among these is their classification of clean and polluted fog episodes. If one examines the back-trajectories, one finds very similar transport patterns in some cases for polluted and non-polluted cases. Furthermore, one can even find sequential samples within a single fog episode that are classified as clean and as polluted. Episode 7 is a good example, where sample 1 is classified as polluted, sample 2 is clean, and sample 3 again polluted. As shown in Figure 7, these samples all have essentially the same transport pattern. It is completely unreasonable to make such a separation based on PM2.5 concentration, especially since the measured PM2.5 in fog does not represent the actual fine particle load upon which the cloud formed since many particles are scavenged in fog and not, therefore, measured by the PM2.5 monitor inside a cloud.

4. Further issues regarding the author's classification of fog samples are apparent in the various attempts to statistically compare bacterial composition across fog samples.

Looking at fog episode 7 again, as one example, one finds samples 1, 2, and 3 end up in very different clusters in Fig. 2. Likewise sequential "clean samples" 1-2 and 1-3 cluster very differently. These observations suggest to me that the author's approach may not be getting at real differences driving bacterial populations.

5. The manuscript lacks adequate description of sampling methodology. One important issue when measuring cloud composition is how the cloud collector is cleaned. This is particularly true for biological sample characterization as attempted here. How was the cloud collector cleaned? Was it sterilized? Was it cleaned just prior to each cloud event? Was the collector kept closed prior to cloud interception to ensure it did not become contaminated? Were cloud collector blanks taken? What bacteria were found in blanks? How do these relate to bacteria observed in samples? Without such information one cannt trust the measured bacteria to have come only from the cloud and not from the sampler.

6. The manuscript is not well written. Grammar and syntax are very poor. At many points the authors' use of English language makes it difficult for the reader to even understand their meaning. Looking closely just at the abstract I counted more than 20 corrections needed to the text and several instances where the authors' meaning was unclear. I did look at some of the manuscript changes recently posted by the authors in response to other reviewer comments and found some improvements to the manuscript text but still observed many problems with the language.

Minor comments:

A. The cloud collector is not properly described. A CASCC2 has a flow rate below 5 m3/min. The 24 m3/min flow rate specified corresponds to a CASCC collector. See collector descirptions and flow rates in Demoz et al. (1996) On the Caltech Active Strand Cloudwater Collectors. Atmos. Res., 41, 47-62.

B. More information needs to be provided about the trajectory calculations. What heights were used as trajectory endpoints?

C. More information should be given about sample handling. The biological samples should have been frozen, not refrigerated at 4 C. How much sample was collected? How much was used in the DNA workup?

D. Some of the fog collection periods were quite long – up to 9 hrs. Was the fog continuously present during this entire period? If not, collected fog water could evaporate and aerosol particles could be captured on collector surfaces, contaminating the fog sample.

E. It would be helpful to include additional information about the fog samples? At a minimum, the authors should include standard parameters such as cloud liquid water content during the sample, concentrations of major ions (which would provide greater insight into pollution levels), and cloud water total organic carbon.

F. The water samples collected atop Mt. Tai in summer are almost certainly associated with intercepted clouds. I suggest the authors not refer to these as fogs.

---

## Author Response (AR1)

**Characteristics of bacterial community in cloud water at Mt. Tai: similarity and disparity under polluted and non-polluted cloud episodes**

Min Wei [1], Caihong Xu [1], Jianmin Chen [1,2,*], Chao Zhu [1], Jiarong Li [1], Ganglin Lv [1]

[1] Environment Research Institute, School of 5 Environmental Science and Engineering, Shandong University, Ji'nan 250100, China

[2] Shanghai Key Laboratory of Atmospheric Particle Pollution and Prevention (LAP), Fudan Tyndall Centre, Department of Environmental Science & Engineering, Fudan University, Shanghai 200433, China

*Correspondence to*: JM.Chen (jmchen@sdu.edu.cn)

**Response to editor**

We thank the editor for the opportunity to respond to reviewer comments. We also thank the reviewers for the beneficial comments on our manuscript. We respond to the reviewer comments in detail below. The responses to reviewer are in red. We also attach a revised manuscript with tracked changes and the amendments were highlighted with yellow color in the revised manuscript.

**Response to reviewer 1**

The authors have performed taxonomic analysis of the metagenomes from polluted and non-polluted episodes of fog events at Mount Tai. The study has described the key differences in the bacterial composition and associated it with the environmental factors. The results are interesting, however, several key points need to be addresses:

**1.** The authors performed a 16S analysis of the microbial community which is an excellent and reliable approach to classify the composition. However, the functionality of microbes cannot be assumed on this base. It would be other approaches (e.g. metagenome-assembly based analysis). Therefore, the authors should modify the text accordingly i.e. by not assuming the functional diversity of fog microbes. Please refer to Jiang et. al, Nat Protoc, 2015 for general methodology.

**Response of the authors:** thank you for your comments.
First, yes, the Miseq 16S rRNA gene sequencing was a powerful tool in microbial diversity investigation, which provides a comprehensive understanding of community composition. The metagenome-assembly based analysis have been widely used in microbial community functional analysis, e.g., Cao et al. (2014) described the microbial communities in $PM_{2.5}$ and $PM_{10}$ using metagenomics during a serious smog event (Cao et al., 2014), Be et al examined aerosolized microorganisms in urban airborne microbes and revealed the metagenomic complexity of urban aerosols and the potential of genomic analytical techniques for biosurveillance and monitoring of threats to public health (Be et al., 2015). We also studied the suggested reference Jiang et. al for general methodology (Jiang et al., 2015). However, due to the

complexity of cloud water collection, the amount for each cloud episode ranged from 40 to 200 mL based on the duration and characteristics of the clouds. The sampled volume was inadequate for metagenomic analysis.

Second, community functions are based on community composition, bacterial taxa. For specific functional bacteria, e.g. Rhizobia (*Phyllobacterium myrsinacearum*) are involved in Biological nitrogen fixation, and favorable for plant growth, Methanotrophic Bacteria (*Methylobacterium aquaticum*, *Methylobacterium adhaesivum*) are related to methane oxidation and carbon recycling. We have added the references attached to the bacterial species in Table 3.

Third, to acquire more accurate community function, we performed PICRUST function prediction in the revised manuscript in section 3.1 (page 11). Phylogenetic Investigation of Communities by Reconstruction of Unobserved States (PICRUST) can be used to predict the metabolic function spectrum of corresponding bacteria and archaea based on the 16S rRNA gene sequence (Langille et al., 2013). PICRUST has been used in bacterial diversity and function analysis (Corrigan et al., 2015; Wu et al., 2016) and it was applied in community function prediction in the present study. The predicted function including metabolism and human disease are essential part of atmospheric microbial community function, which are likely attributed to the bacterial gene content from the identified species in Table 3.

[Figure]

**Predictive functional profiling of microbial communities, using 16s rRNA gene sequences**

**2.** Quality of English needs to be revised. Grammatical mistakes and use of wrong terminologies is seen throughout the manuscript. For instance, grammatical mistake

in introduction (Line 5), Page 9 line 33 (": : : server infections") and Page 10 line 5 ("Previous studies has shown : : :"). The author names also have discrepancy in main text and supplement.

**Response of the authors:** We have polished the manuscript with a professional assistance in writing. The mistakes have been corrected.

**3.** Few major statements are not supported by references e.g. the authors mentioned that the recruitment of bacteria from various sources to air is harmful to humans upon deposition back to land via fog.

**Response of the authors:** Numerous studies are focused on the potential pathogens identified in atmospheric particulate matter ($PM_{2.5}$ and $PM_{10}$) (Cao et al., 2014; Creamean et al., 2013), rain water (Kaushik & Balasubramanian, 2012; Simmons et al., 2001), and indicated that health risk-related bacteria in atmospheric samples should be concerned. For cloud water, studies of health risks to individuals are typically focused on the chemical characteristic, e.g. the low pH (Hackney et al., 1989), PAH (Ehrenhauser et al., 2012), etc. Limited literatures discussed the microorganism in cloud water suggested the potential pathogens in cloud water (Vaïtilingom et al., 2012), they find potential plant pathogens such as *Pseudomonas syringae* and *Xanthomonas campestris*. In their study, they suggest that the wet deposition play a major role in the dispersion of microorganisms, which appear as an extension of the phyllosphere and carry living species of plant pathogens that could then infect new hosts through precipitation.
In the present study, potential human pathogens were identified in the cloud water samples. However, the detailed health risks should be prudently assessed. Further study depending on the culture-dependent method and biochemical experiments will perform to check the pathogenicity. We have revised the relevant discussion and added reference in page 12 (from line 25 to line 33), page 14 (from line 27 to line 35) and page 15 (from line 1 to line 14).

**4.** Figure captions need to a bit more descriptive/proof read.

**Response of the authors:** More detailed descriptions have been added to the figure captions.
**Figure 1** Bacterial community variation for fog episodes at the phylum (A) and class (B) level. FE refers to the cloud episodes. Bar graphs for each sample represent the percentage of taxa assigned to each phylum with 80% bootstrap confidence. Taxonomic summary of the most abundant taxa (more than 1%) across all cloud samples are indicated in the figure.
**Figure 2** Bacterial taxa are related to KEGG functional pathways. Bacterial gene functions were predicted from 16S rRNA gene-based microbial compositions using the PICRUSt algorithm to make inferences from KEGG annotated databases. Spearman's correlation coefficients were estimated for each pairwise comparison of

genus counts and KEGG pathway counts. Selected KEGG pathways relating to metabolism and disease infection and predominant genera are included in the heatmap. Red color refers to the positive correlation, and green indicates a negative correlation. Correlation is significant at $*P < 0.05$, $**P < 0.01$.

**Figure 3** Schematic representation of bioaerosols life cycle and potential influence on atmosphere, ecosystem and human health, modified from Poeschl (Poeschl, 2006). The predominant bacteria species with potential functions are indicated in the figure. Bioaerosols emissions and resuspension from various terrestrial environments, e.g., soil, water, plants, animals or human beings, may include pathogenic or functional species. These bacteria can be attached to particles or incorporated into water droplets of clouds/fog. Certain species can serve as biogenic nuclei for Cloud Condensation Nuclei (CCN) and Ice Nuclei (IN), which induce rain formation, precipitation, and wet deposition of gases and particles. For the potential pathogens and functional bacteria, during cloud process, they can be deposited back to land via deposition and possibly induce infections to human health and impose effect on the diversity and function of aquatic and terrestrial ecosystems.

**Figure 4** Bacterial taxa significantly differentiated between the polluted and non-polluted cloud episodes identified by linear discriminant analysis coupled with effect size (LEfSe). The LDA effect sizes (left) were calculated using the default parameters. The taxonomic cladogram (right) with LDA values higher than 3.5 comparing all bacterial taxa and significantly discriminant taxon nodes are colored and branch areas are shaded according to the highest-ranked variety for that taxon. Taxa with significant difference in polluted and non-polluted cloud episodes are indicated in red and green circles, respectively. The bacterial taxa with nonsignificant differences are represented as yellow circles and the diameter of the circles are proportional to relative abundance. The abbreviation in the cladogram tree: a: g_Rhodococcus, b: f_Nocardiaceae, c: f_Flavobacteriaceae, d: o_Flavobacteriales, e: c_Flavobacteriia, f: p_Bacteroidetes, g: f_Deinococcaceae, h: o_Deinococcales, i: c_Deinococci, j: p_Deinococcus-Thermus, k: f_Bacillaceae, l: o_Clostridiales, m: c_Clostridia, n: p_Firmicutes, o: f_Oxalobacteraceae, p: o_Burkholderiales, q: c_Betaproteobacteria, r: f_Moraxellaceae, s: o_Pseudomonadales, t: f_Xanthomonadaceae, u: o_Xanthomonadales, v: c_Gammaproteobacteria, w: p_Proteobacteria.

**Figure 5** Biplot of the environmental variables and genus-level community structure using a redundancy analysis (RDA) model, describing the variation in bacterial community explained by environmental variable. CE refers to cloud episodes. Polluted episodes are indicated in red circle, and non-polluted episodes are green squares. Species data are listed in Table S2. The selected environmental variables are significant ($P < 0.05$) using Monte Carlo permutation testing. Species are labeled with triangle, the closer of which indicates existence in similar environment. Environmental variables are showed by arrows; the relative length is positive correlation with the importance in influencing bacterial community structure. The angle between the arrow and the ordination axis suggest the variable response respect to the RDA gradient. The two axes explain 65.9% of the variability. For bacteria,

major ions in cloud water seem to be the most important environmental variable shaping the community structure.

**Figure 6** Air mass transport pathways for the cloud episodes using the Hybrid Single Particle Lagrangian Integrated Trajectory (HYSPLIT) model. 24-hour backward trajectories were calculated for air parcels arriving at the summit of Mt. Tai (36o18′ N, 117o13′ E, and 1534 m a.s.l). CE refer to cloud episodes. The polluted episodes are indicated in red lines, and green lines are non-polluted episodes.

**Figure 7** Wind Rose Diagram to quantitative analysis of wind speed and wind direction during sampling time between polluted (A) and non-polluted cloud episodes (B). The frequency of winds is plotted by wind direction, the color bands show wind speed range. The direction of the longest spoke shows the wind direction with the greatest frequency. Figure 7 (C) indicates distribution of wind speed during sampling time and correlation with $PM_{2.5}$ concentration. As shown in the figure, $PM_{2.5}$ concentration was high under lower wind speed, whereas $PM_{2.5}$ was lower when wind speed was high.

**Response to reviewer 2**

The authors investigated the differences in bacterial community structures from fog water droplet samples collected from Mt. Tai in North Plain of China including those clear and polluted days in July and August of 2014. They performed sequence analysis of the samples, and also investigated the effects of environmental factors on the bacterial community structure. Overall, it is interesting to study the bacteria in the fog water samples, especially in higher altitude from a ground. The information developed is useful to understanding the microbial transport and possible roles in atmospheric pollutant transformation. The authors provided a number of different analyses of their results and derived some valuable information. Nonetheless, this reviewer does observe the following drawbacks that need the authors' attention:

**1.** From their work, it seems they only had one day with higher PM2.5 pollution level, i.e., exceeding 100 ug/m^3, and they had more samples from clear days with much lower PM2.5 levels. In their work, they compared them and further derived relevant information. I think the authors have to carefully make their conclusions regarding their limited set of data from a single polluted day. Probably, they can use the 24-hour backward trajectories to discuss more on them.

**Response of the authors:** According to our field observation data on the summit of Mt.Tai in the year of 2014 and 2015(unpublished), the 24 h average $PM_{2.5}$ mass concentration was basically less than 100 μg/m3, sometimes with relative lower concentration less than 10 μg/m$^3$. The possible reason was that the Mt. Tai was the highest mountain in North China Plain (1534 m a.s.l), which was typically as the background of atmospheric quality. The $PM_{2.5}$ was relatively low than other regions in the North China Plain. Similar results were obtained by other studies and suggest the relation of $PM_{2.5}$ and attitude. Gehrig and Buchmann studied the seasonal

variations and spatial distribution of ambient $PM_{10}$ and $PM_{2.5}$ concentrations. In comparison to other study area (different attitude), the lowest $PM_{2.5}$ concentrations were observed at the elevated site Chaumont (1140 m a.s.l.) (Gehrig & Buchmann, 2003). Similarly, Fan et al studied the vertical distribution of $PM_{2.5}$ concentration in fog and haze days in Beijing and suggest that $PM_{2.5}$ concentrations decreased with the increase of altitude (Fan et al., 2009).

**The 24 h average $PM_{2.5}$ mass concentration according to our field observation data**

| 2014 | PM2.5 ($\mu g/m^3$) | 2014 | PM2.5 ($\mu g/m^3$) | 2015 | PM2.5 ($\mu g/m^3$) | 2015 | PM2.5 ($\mu g/m^3$) |
|---|---|---|---|---|---|---|---|
| 2014/7/23 | 53.9 | 2014/8/9 | 30.1 | 2015/7/6 | 62.3 | 2015/7/23 | 34.8 |
| 2014/7/24 | 49.9 | 2014/8/10 | 67.5 | 2015/7/7 | 64.8 | 2015/7/24 | 34.2 |
| 2014/7/25 | 12.2 | 2014/8/11 | 51.1 | 2015/7/8 | 112 | 2015/7/25 | 36 |
| 2014/7/26 | 44.5 | 2014/8/12 | 45.5 | 2015/7/9 | 71.6 | 2015/7/26 | 33.7 |
| 2014/7/27 | 66.9 | 2014/8/13 | 47.4 | 2015/7/10 | 61.2 | 2015/7/27 | 65.6 |
| 2014/7/28 | 97.5 | 2014/8/14 | 49.3 | 2015/7/11 | 80.6 | 2015/7/28 | 49 |
| 2014/7/29 | 73.4 | 2014/8/15 | 47.6 | 2015/7/12 | 31.4 | 2015/7/29 | 7.9 |
| 2014/7/30 | 23.4 | 2014/8/16 | 66.2 | 2015/7/13 | 57 | 2015/7/30 | 12.6 |
| 2014/7/31 | 56.2 | 2014/8/17 | 69.5 | 2015/7/14 | 53.5 | 2015/7/31 | 14.9 |
| 2014/8/1 | 17.7 | 2014/8/18 | 53.7 | 2015/7/15 | 52.1 | 2015/8/1 | 17.2 |
| 2014/8/2 | 42.5 | 2014/8/19 | 64.9 | 2015/7/16 | 54.5 | 2015/8/2 | 35 |
| 2014/8/3 | 45.5 | 2014/8/20 | 62.1 | 2015/7/17 | 62.9 | 2015/8/3 | 6.1 |
| 2014/8/4 | 86.5 | 2014/8/21 | 71.3 | 2015/7/18 | 51.9 | 2015/8/4 | 5.9 |
| 2014/8/5 | 54.8 | 2014/8/22 | 54.2 | 2015/7/19 | 44.4 | 2015/8/5 | 9.9 |
| 2014/8/6 | 18.2 | 2014/8/23 | 48.8 | 2015/7/20 | 21.4 | 2015/8/6 | 19.3 |
| 2014/8/7 | 44.1 | 2014/8/23 | 42.8 | 2015/7/21 | 43.1 | 2015/8/7 | 7.8 |
| 2014/8/8 | 40.3 | | | 2015/7/22 | 55.8 | 2015/8/8 | 26.2 |

In addition, the listed $PM_{2.5}$ concentration in Table 1 was the average value during a cloud process, not the 24 h average concentration. The 24 h $PM_{2.5}$ concentration in cloud days was lower than non-cloud days which possible due to the wet deposition. During cloud episodes, $PM_{2.5}$ concentration varied with cloud process. The mass concentration was high in the initiation of cloud episode, with the development and dissipation of cloud, the concentration steadily reduced due to the reduced input (nighttime) and wet deposition.

In the present study, the polluted cloud episodes were firstly defined according to the 24 h concentration of WHO air quality guideline (25 $\mu g/m^3$) and the standard was applied by Australia, New Zealand and European Union. In the revised manuscript, we checked the major ions in cloud water and reclassify the cloud episodes according to the concentration of major ions in cloud water. Cloud episodes under high concentration of ions in cloud water and high atmospheric $PM_{2.5}$ concentration were defined as polluted. The cluster and PCA analysis also confirmed the classification.

In the section of 3.4, we have discussed the influence of air mass and meteorological conditions on $PM_{2.5}$. The Sampling site was 1534 m a.s.l, air pollution was typically effected by air mass over long term transport than local emissions.We use the 24-hour backward trajectories to track the air mass and combined the wind direction and wind

speed to deeply discuss the possible driven factors. The main points obtained was that air mass from the contaminated area through long term transport with lower wind speeds, largely reduce the diffusion rate of pollutants and thus lead to the sustained high $PM_{2.5}$ during polluted cloud episodes.

**2.** It seems they did not clearly define what level of PM2.5 for which a day can be classified as a polluted day in their method section. Also they should clearly define what those symbols such as "FE" stand for? although I guess it should be "Fog Episode", but they should appear in all figure captions so that readers can easily understand the figures. They should describe that the characteristics of each Fog Episode are shown in relevant Tables in each Figure.

**Response of the authors:** thank you to your suggestion, we have replaced "fog episodes" as "cloud episodes" and described cloud episode in relevant Tables and Figures.
We have clearly defined the polluted cloud episodes in page 5 (from line 21 to line 29), page 9 (from line 14 to line 26) and indicated the abbreviation in all the table and figures. The polluted cloud episodes were firstly defined according to the 24 h concentration of WHO air quality guideline (25 μg/m$^3$) and the standard was applied by Australia, New Zealand and European Union. During a cloud episode, the average $PM_{2.5}$ concentration higher than 25 μg/m$^3$ was classified as polluted. WHO proposes $PM_{2.5}$ less than 10 ug/m$^3$ is safe. Elevated $PM_{2.5}$ concentration will highly increase health risks. The high pollutant and pathogens are detrimental to individuals (Fang et al., 2013). $PM_{2.5}$ concentrations were compared to the 24 h World Health Organization limit of 25 μg/m$^3$.
In the revised manuscript, we checked the major ions in cloud water and reclassify the cloud episodes according to the concentration of major ions in cloud water. Cloud episodes under high concentration of ions in cloud water and high atmospheric $PM_{2.5}$ concentration were defined as polluted. The cluster and PCA analysis also confirmed the classification.

**3.** be aware that they only performed genus level sequence and they cannot derive any particular bacterial species, especially when they discuss about pathogens. For certain genera, not all of their species are pathogens or opportunistic pathogens.

**Response of the authors:** we have revised the discussion about potential pathogens in page 12 (from line 25 to line 33), page 14 (from line 27 to line 35) and page 15 (from line 1 to line 14). Yes, the Miseq sequencing can identify bacterial taxa mostly at the genus level. In the present study, the V3-V4 hypervariable region of the bacterial 16S rRNA gene was amplified. Single OTUs were removed and taxonomy was assigned to each OTU using the Ribosomal Database Project (RDP) classifier in QIIME, with a minimum confidence cutoff of 80% against the Silva reference database (silva 119,http://www.arb-silva.de/) to the genus level. Finally, we focused on those bacterial genera that included species known or suspected to be opportunistic

pathogens. To this aim, we performed a systematic literature review to identify potential pathogenic bacteria in water habitats (Bibby et al., 2010; Guo & Zhang, 2012; Luo & Angelidaki, 2014).

Previous studies have discussed the bacterial pathogens based on NGS sequencing (454 pyrosequencing, Miseq, Ion Torrent PGM). Razzauti et al conducted a comparison between transcriptome sequencing and 16S metagenomics for detection of bacterial pathogens in Wildlife (Razzauti et al., 2015) and suggest that 16S approach was able to determine bacterial diversity in each individual. They also indicated that NGS techniques (454-pyrosequencing and MiSeq) are very affordable candidates and could become routine approaches in future large-scale epidemiological studies. Luo and Angelidaki studied the bacterial communities and bacterial pathogens with the high sequencing depth by Ion Torrent PGM (16S rRNA gene sequencing), they suggest the Ion Torrent PGM is also possible to detect the potential bacterial pathogens in biogas reactors. To identify potential pathogens, they use the reference bacterial pathogen database and identified the potential bacterial pathogens at the species level (Luo & Angelidaki, 2014).

**4.** I did not see any concentration levels for the total bacteria in their fog water droplet samples? Did they perform qPCR for total bacteria for their samples?

**Response of the authors:** Due to the complexity of cloud water collection, the amount for each cloud episode ranged from 40 to 200 mL based on the cloud duration and characteristics. For the majority episodes, e.g. CE1-3, CE2-1, CE4-1, CE4-2, the remained volume was inadequate for other analysis after Miseq sequencing.

The collected cloud water samples were processed by genomic DNA extracting, PCR amplification, Miseq sequencing and qPCR. In DNA extraction, some samples DNA cannot be successfully extracted and require repeated extraction, thus consume more sample volume. We have performed qPCR for total bacteria after Miseq. However, after miseq, no remaining sample DNA for the further analysis for certain samples. QPCR was just performed for the samples with sufficient DNA and bacterial concentration are listed in the following table. Therefore, we did not discuss the total bacterial concentration in the manuscript.

Bacterial concentration for different cloud episodes

| Sample | Collected volume (mL) | Bacterial concentration (cells/mL) |
|---|---|---|
| CE1-1 | 90 | $8.9 \times 10^4$ |
| CE1-2 | 80 | $1.3 \times 10^5$ |
| CE1-3 | 55 | Not detected |
| CE2-1 | 75 | Not detected |
| CE3-1 | 100 | Not detected |
| CE4-1 | 65 | Not detected |
| CE4-2 | 40 | Not detected |
| CE4-3 | 40 | Not detected |
| CE5-1 | 50 | Not detected |

| CE6-1 | 60 | Not detected |
|---|---|---|
| CE7-1 | 210 | $1.5 \times 10^5$ |
| CE7-2 | 200 | $5.8 \times 10^4$ |
| CE7-3 | 120 | $1.6 \times 10^5$ |

**5.** It would be great if they can provide data for fungal spores. I guess there will be some fungal spores in the fog water droplets.

**Response of the authors:** We agree that the investigations of fungal diversity in fog/cloud water are very important areas for future work. Your suggestions are very helpful for our further study. However, the analysis of fungal spores requires substantial amounts of additional work, including the resequencing and culture-dependent experiments. The remaining parts of the samples are unable to support the above experiments. We therefore decide not to include these in our current manuscript and leave for future work. The next studies on microbial community will consider the fungal diversity in fog/cloud water and other aerosol samples.

**6.** For certain bacteria, when they are stored at 4 degree C, they can still grow. How long did it elapse between the collection and their actual analysis?

**Response of the authors:** thank you for your comments. We have modified the description and clearly described the storage conditions of the sample in page 5(line 32) and page 6 (from line 1 to line 2). Basal analysis of water typically included chemical and biological two parts. For chemical analysis, part of samples were stored in pre-baked glass bottles, immediately preserved with hydrochloric acid (HCl, pH <2.0), stored at 4 ℃ in ice box during transit, and analyzed upon arrival at the Laboratory. Samples for microbial diversity analysis were not preserved with hydrochloric acid and stored with dry ice in transit, and frozen at -80 ℃ until further analysis.

**Response to reviewer 3**

Min et al examine bacteria present in cloud water samples collected at Mt. Tai, China. They use a variety of techniques to examine the community composition of bacteria in the samples and attempt to assess differences as a function of a variety of environmental parameters, especially fine particle concentration levels. While the dataset is interesting and the work novel, I have numerous concerns about the work and its presentation.

Major comments:
1. The authors never make it very clear why they are examining bacteria in clouds (they are looking at clouds, not fog – see below). They talk about the importance of interaction with fog, but don't clarify why such interactions are important. They speak about deposition in clouds, but why is this really important if such bacteria would be

deposited anyway by wet or dry processes? Bacteria in cloud drops get there through scavenging of aerosol particles that are either themselves bacteria or have bacteria attached. Why, then, is it important to look at bacteria in cloud water? Why not look at them directly in PM2.5 samples? This would allow a much larger dataset to be examined, which would greatly help statistical analyses of relationships with environmental variables. For example, if one is interested in examining changes in bacterial populations with PM2.5 levels, it would be much more straightforward to look at bacteria directly in PM2.5.

**Response of the authors:** According to your suggestion, we have revised the manuscript in the introduction (page 3, line 29-32; page 4, line 1-15) and defined the samples as clouds.

In recent years, Northern China experienced serious air pollution. Mt. Tai ($36^o18'$N, $117^o13'$E, and 1534 m a.s.l), locates on the summit of North China Plain, is frequently attacked by cloud events (Guo et al., 2012; Liu et al., 2012; Wang et al., 2011). In contaminated area, cloud typically contains numerous pollutants, e.g., sulfate and nitrate ions, organic carbon compounds, and bioaerosols. During cloud process, atmospheric bacteria attached to particles or incorporated in cloud droplets will be deposited back to land via wet deposition, which may induce health risks through microbial pathogens dispersion and potential effect on the diversity and function of aquatic or terrestrial ecosystems. Previous literature discussed the microorganism in cloud water suggested the potential pathogens in cloud water (Vaïtilingom et al., 2012). They find potential plant pathogens such as *Pseudomonas syringae* and *Xanthomonas campestris*. They also suggest that the wet deposition play a major role in the dispersion of microorganisms, which could then infect new hosts through precipitation. Therefore, to evaluate the potential ecological functional bacteria in cloud water have been an urgent issue, especially for the polluted cloud episodes.

In cloud, bacteria can act as efficient cloud condensation nuclei or ice nuclei and associate with biogeochemical cycling (nitrogen/carbon cycling), which have been demonstrated by numerous studies (Amato et al., 2015; Hill et al., 2007; Vaïtilingom et al., 2013). Similarity, previous literatures have studied the bacterial community in rain or snow (Cho & Jang, 2014; Mortazavi et al., 2015). They also focus on the bacteria associated with CNN/IN, potential pathogens and biochemical reactions. In cloud, bacterial activities were essential to atmospheric hydrogenic cycle. To understand the bacterial community structure was the first step for further study on their activities in cloud. Therefore, we should first investigate bacterial community in cloud.

In the revised manuscript, we discussed the relationship between $PM_{2.5}$, major ions and bacteria. Major ions in cloud have vital role in bacterial community structure variation. In our opinion, major ions have direct influence on bacterial community, whereas $PM_{2.5}$ had an indirect impact on bacterial community structure.

2. One might be interested in examining how cloud processing affects bacteria. For example, do they differentially scavenge and deposit bacteria from a certain subset of

aerosol particles? Do the bacteria reproduce in clouds as suggested by Fuzzi? Does interaction with fogs alter the viability of bacteria in some way. The authors do not examine any such questions that would be very relevant to bacteria in fog.

**Response of the authors:** Your suggestions are very helpful for our further research. All the suggested researches are based on the in situ analysis during sampling and culture-dependent experiments in lab. We will perform the culture-dependent methods in laboratory to check the bacterial activity in further study, such as isolation of viable bacteria under low temperature and drop-freezing assays.
In the present study, we tell the readers which bacteria were in cloud water and whether there was different between polluted and non-polluted cloud by investigation of community structure. The basic study will provide fundamental acquaintance for the further research on bacterial activity in cloud.

3. I have many concerns about the way in which the authors assess differences in bacteria in fog between polluted and nonpolluted conditions. Chief among these is their classification of clean and polluted fog episodes. If one examines the back-trajectories, one finds very similar transport patterns in some cases for polluted and non-polluted cases. Furthermore, one can even find sequential samples within a single fog episode that are classified as clean and as polluted. Episode 7 is a good example, where sample 1 is classified as polluted, sample 2 is clean, and sample 3 again polluted. As shown in Figure 7, these samples all have essentially the same transport pattern. It is completely unreasonable to make such a separation based on $PM_{2.5}$ concentration, especially since the measured $PM_{2.5}$ in fog does not represent the actual fine particle load upon which the cloud formed since many particles are scavenged in fog and not, therefore, measured by the $PM_{2.5}$ monitor inside a cloud.

**Response of the authors:** we have redefined the polluted and non-polluted cloud episodes in the revised manuscript. Polluted cloud episodes were defined based on concentration of $PM_{2.5}$ and major ions in cloud water not just based on $PM_{2.5}$ concentration.
In continues cloud episodes, back-trajectories may be similar transport patterns. However, $PM_{2.5}$ concentration varied during cloud process. We measured the atmospheric $PM_{2.5}$ concentration and the variation of $PM_{2.5}$ was caused by wet deposition during cloud process. In the cloud formation stage, $PM_{2.5}$ concentration was high. With the development, the soluble components in $PM_{2.5}$ were scavenged in cloud. Atmospheric $PM_{2.5}$ gradually decreased. However, new pollutant from local regional emission will cause the elevated $PM_{2.5}$ (see cloud episode 7). For instance, in cloud episode 7, $PM_{2.5}$ concentration increased in the dissipation stage from 6:00 to 9:00 in the morning. The busy traffic vehicle and the industrial and agricultural activities resulted in the emissions of new pollutants into atmosphere. Therefore, an increased $PM_{2.5}$ concentration was observed.
In the revised manuscript, we have defined the polluted and non-polluted cloud episodes based on the major ions and $PM_{2.5}$ concentration. Cloud episodes under high

concentration of ions in cloud water and high atmospheric PM$_{2.5}$ concentration were defined as polluted.

[Figure]

| | CE7-1 | CE7-2 | CE7-3 |
|---|---|---|---|
| Time | 2:30-4:38 | 4:38-6:21 | 6:21-9:20 |
| PM$_{2.5}$($\mu$g·m$^{-3}$) | 30.45 | 23.39 | 41.60 |
| EC ($\mu$S·cm$^{-1}$) | 356.20 | 207.50 | 187.60 |
| Cl$^-$($\mu$eq$^{-1}$) | 182.61 | 47.50 | 83.92 |
| NO$_3^-$($\mu$eq$^{-1}$) | 581.20 | 281.20 | 330.74 |
| SO$_4^{2-}$($\mu$eq$^{-1}$) | 2391.69 | 1045.64 | 1633.34 |
| Na$^+$($\mu$eq$^{-1}$) | 70.42 | 31.33 | 38.40 |
| NH$_4^+$($\mu$eq$^{-1}$) | 2599.56 | 1721.15 | 2412.55 |
| K$^+$($\mu$eq$^{-1}$) | 85.77 | 42.20 | 61.15 |
| Mg$^{2+}$($\mu$eq$^{-1}$) | 156.02 | 72.57 | 68.03 |
| Ca$^{2+}$($\mu$eq$^{-1}$) | 963.70 | 417.79 | 387.14 |

4. Further issues regarding the author's classification of fog samples are apparent in the various attempts to statistically compare bacterial composition across fog samples. Looking at fog episode 7 again, as one example, one finds samples 1, 2, and 3 end up in very different clusters in Fig. 2. Likewise sequential "clean samples" 1-2 and 1-3 cluster very differently. These observations suggest to me that the author's approach may not be getting at real differences driving bacterial populations.

**Response of the authors:** The classification of fog samples were first based on the 24 h concentration of WHO air quality guideline (25 $\mu$g/m$^3$) and this standard has been applied by Australia, New Zealand and European Union. During a fog episode, the average PM$_{2.5}$ concentration higher than 25 $\mu$g/m$^3$ was classified as polluted.
In the revised manuscript, we checked the major ions in cloud water and reclassify the cloud episodes according to the concentration of major ions in cloud water.
For cloud episode 7, as mentioned in question 3, PM$_{2.5}$ and major ions dymaic with cloud process. In the cloud formation stage, PM$_{2.5}$ concentration was high. With the development of cloud, the soluble components in PM$_{2.5}$ were scavenged in cloud. Atmospheric PM$_{2.5}$ gradually decreased. However, new pollutant from local regional emission will cause the elevated PM$_{2.5}$ (see cloud episode 7). The varied PM$_{2.5}$ and major ions concentration were responsible for the different distribution. The concentration of major ions in the samples of CE7-1 and CE7-3 were higher than in CE7-2. Sample CE7-2 was cleaned with lower PM$_{2.5}$ and ions concentration.
Although the PM$_{2.5}$ concentration for sample CE1-2 was low, a relative high major ions concentration was detected in cloud water. Therefore, we categorized the sample CE1-2 as polluted sample. The cluster and PCA analysis also confirmed the classification (Figure S4), CE1-2 was closely to other polluted samples.

5. The manuscript lacks adequate description of sampling methodology. One important issue when measuring cloud composition is how the cloud collector is cleaned. This is particularly true for biological sample characterization as attempted here. How was the cloud collector cleaned? Was it sterilized? Was it cleaned just prior to each cloud event? Was the collector kept closed prior to cloud interception to ensure it did not become contaminated? Were cloud collector blanks taken? What bacteria were found in blanks? How do these relate to bacteria observed in samples? Without such information one cannot trust the measured bacteria to have come only from the cloud and not from the sampler.

**Response of the authors:** we have added the description of sampling methodology in page 5(from line 8 to line 20). The cloud collector was cleaned prior to each cloud event and kept closed prior to cloud interception to ensure not to be contaminated. The collector was activated by a sensor only when cloud formed in the ambient air. The cloud water was aspirated through a Teflon duct at a rate of 24.5 $m^3$ $min^{-1}$ by a fan situated at the rear of Caltech Active Strand Cloud water Collector (CASCC) (See the attached figure). Cloud water collected from the Teflon strands, through Teflon tube and down into Teflon bottles. The theoretical 50% cut-off size was equivalently drop diameter of 3.5 μm.

To avoid the artificial and instrumental contamination, the Teflon tube and the polyethylene bottles were first pretreated with anhydrous ethanol and then washed 3 times using the sterilized ultrapure water. Before sampling, the cloud collector was washed with the sterilized deionized distilled water filtered through the with 0.22 μm membrane. Then spray the sterilized dd-$H_2O$ into the collector using the sprayer and the collected water sample was as the blank. The parallel aliquots of sterile dd-$H_2O$ were run through an identical DNA extraction procedure to check for sample contamination. These DNA extraction "blanks" were PCR amplified alongside the DNA samples extracted from the cloud water samples. Genomic DNA cannot be extracted from the blank samples. Therefore, we considered the cloud collector was strictly sterilized and cleaned before sampling and cannot contaminate the collected cloud water.

6. The manuscript is not well written. Grammar and syntax are very poor. At many points the authors' use of English language makes it difficult for the reader to even understand their meaning. Looking closely just at the abstract I counted more than 20 corrections needed to the text and several instances where the authors' meaning was unclear. I did look at some of the manuscript changes recently posted by the authors in response to other reviewer comments and found some improvements to the manuscript text but still observed many problems with the language.
**Response of the authors:** We have polished the manuscript.

Minor comments:
A. The cloud collector is not properly described. A CASCC2 has a flow rate below 5 m3/min. The 24 m3/min flow rate specified corresponds to a CASCC collector. See

collector descirptions and flow rates in Demoz et al. (1996) On the Caltech Active Strand Cloudwater Collectors. Atmos. Res., 41, 47-62.

**Response of the authors:** The CASCC collector was used during cloud water collecction, we have corrected.

B. More information needs to be provided about the trajectory calculations. What heights were used as trajectory endpoints?

**Response of the authors:** We have indicated the location and heights in figure. The height for the backward trajectories was 1534 m (the height of sampling site on the summit of Mt. Tai).

C. More information should be given about sample handling. The biological samples should have been frozen, not refrigerated at 4 C. How much sample was collected? How much was used in the DNA workup?

**Response of the authors:** We have answered the question in the response to review 1. Basal analysis of water typically included chemical and biological two parts. For chemical analysis, part of samples were stored in pre-baked glass bottles, immediately preserved with hydrochloric acid (HCl, pH <2.0), stored at 4 ℃ in ice box during transit, and analyzed upon arrival at the Laboratory. Samples for microbial diversity analysis were not preserved with hydrochloric acid and stored with dry ice in transit, and frozen at -80 ℃ until further analysis.
The collected sample volume has been listed in the Table (response to reviewer 2, question 4). The collected volume for each sample was varied across the 13 cloud water samples and ranged from 40 to 200 mL. For each sample, at least 20 mL cloud water sample was used in DNA workup.

D. Some of the fog collection periods were quite long – up to 9 hrs. Was the fog continuously present during this entire period? If not, collected fog water could evaporate and aerosol particles could be captured on collector surfaces, contaminating the fog sample.

**Response of the authors:** The collection of cloud water was continuously during the entire period. As shown in the following figure, the polyethylene bottle was tight with the lid to avoid evaporation and external contamination. Cloud water in air was aspirated through the duct. To avoid the contamination from the aerosol particles and rainwater, a triangular roof was in the front of the duct.

[Figure]

E. It would be helpful to include additional information about the fog samples? At a minimum, the authors should include standard parameters such as cloud liquid water content during the sample, concentrations of major ions (which would provide greater insight into pollution levels), and cloud water total organic carbon.

**Response of the authors:** we have added the data of cloud liquid water content, the concentration of major ions and total organic carbon in cloud water in the Supplement Materials (Figure S1). We also analyzed the correlation between these environmental parameters and bacterial community in the manuscript in the discussion section about bacterial community and environmental factors.

F. The water samples collected atop Mt. Tai in summer are almost certainly associated with intercepted clouds. I suggest the authors not refer to these as fogs.

**Response of the authors:** we have revised the description and replaced "fog" as "cloud".

[revised manuscript text omitted]

Abbreviates are as followed: AP, Alphaproteobacteria; BP, Betaproteobacteria; GP, Gammaproteobacteria; AC, Actinobacteria; BA, Bacteroidetes; CY, Cyanobacteria; DT, Deinococcus-Thermus; FR, Firmicutes. Biodegradation refers to the bacteria associated with the biodegradation of organic compounds, even toxic pollutants, e.g., aromatic compounds.

---

## Author Response (AR2)

**Response to editor**

We thank the editor for the opportunity to respond to reviewer comments. We also thank the reviewers for the comments on our manuscript. We respond to the reviewer comments in detail below. The responses to reviewer are in red. We also attach a revised manuscript with tracked changes and the amendments were highlighted with yellow color in the revised manuscript.

**Response to reviewer 1**

1. The authors were required to make several key changes in their manuscript to improve its quality and clarity. The revised manuscript includes several key references as well analysis of 16S rRNA data using PICRUST to provide more accurate insight of the function of the studied community. The initial submission required significant improvement in quality of the language, which has been taken care of in the revised submission. The authors have now also provided supporting references for several major scientific statements. Furthermore, the figure legends in the previous manuscript did not provide sufficient insight of their respective figures but improved in the revised version. Considering the revision work carried out by the authors and its impact on the improvement of manuscript I now recommend it for publication in the upcoming issue of the journal.

Response of the authors: thank you for your comments. We have revised our manuscript according to the reviewer's suggestions.

**Response to reviewer 2**

1. My only concern for the paper is that the authors should indicate the limitation of their work due to their small sample size. This should be indicated both in the abstract and also in the conclusion section.

Response of the authors: we have indicated the limitation of the work due to small sample size in the abstract and conclusion sections.

In the abstract section:

However, due to the limited sample size (13 samples) collected at the summit of Mt. Tai, these issues

need in-depth discussion. Further studies based on an annual series of field observation experiments and laboratory simulations will continue to track these issues.

In the conclusion section:

However, due to limited sampling size and collected volume, the aforementioned focus needs further discussion. Continuous annual observation and culture-dependent experiments will be performed to target the detailed functions of the atmospheric bacterial community.

2. Besides, the language might need to be further improved, e.g., I do not understand the sentence "... $PM_{2.5}$ seams..." It seems that the authors meant that "$PM_{2.5}$ seems to be a pivotal variable ....?"

Response of the authors: thank you for your comments. We have polished the language, and improved the grammar and expressions.

The sentence has been revised in line 9-10 page 17:

[revised manuscript text omitted]